# Anterior cingulate and medial prefrontal cortex oscillations underlie learning alterations in trait anxiety in humans

Thomas P. Hein[1,4], Zheng Gong[2], Marina Ivanova [2], Tommaso Fedele[2], Vadim Nikulin[3] & Maria Herrojo Ruiz [1,4✉]

Anxiety has been linked to altered belief formation and uncertainty estimation, impacting learning. Identifying the neural processes underlying these changes is important for understanding brain pathology. Here, we show that oscillatory activity in the medial prefrontal, anterior cingulate and orbitofrontal cortex (mPFC, ACC, OFC) explains anxiety-related learning alterations. In a magnetoencephalography experiment, two groups of human participants pre-screened with high and low trait anxiety (HTA, LTA: 39) performed a probabilistic reward-based learning task. HTA undermined learning through an overestimation of volatility, leading to faster belief updating, more stochastic decisions and pronounced lose-shift tendencies. On a neural level, we observed increased gamma activity in the ACC, dmPFC, and OFC during encoding of precision-weighted prediction errors in HTA, accompanied by suppressed ACC alpha/beta activity. Our findings support the association between altered learning and belief updating in anxiety and changes in gamma and alpha/beta activity in the ACC, dmPFC, and OFC.

[1] Goldsmiths, University of London, Psychology Department, Whitehead Building New Cross, London SE14 6NW, UK. [2] Centre for Cognition and Decision making, Institute for Cognitive Neuroscience, HSE University, Moscow, Russian Federation. [3] Department of Neurology, Max Planck Institute for Human Cognitive and Brain Sciences, Leipzig, Germany. [4] These authors contributed equally: Thomas P. Hein, Maria Herrojo Ruiz. ✉email: M.Herrojo-Ruiz@gold.ac.uk

Anxiety is a psychological, physiological, and behavioural state characterised by worry about undetermined events with potentially adverse outcomes[1–3]. A central feature in clinical and subclinical anxiety is difficulty dealing with uncertainty, playing a role in diagnosis and treatment[4–7] as well as in the modelling of anxious responses[8–10]. Computational modelling work has revealed that anxiety impairs learning and decision making when the associations between responses and their outcomes change due to environmental uncertainty or volatility[11–14]. Misestimation of other forms of uncertainty can also account for attenuated learning in anxiety, as shown in temporary anxiety states and in the somatic (physiological) component of trait anxiety[14–16]. These empirical findings converge with proposals that associate affective disorders with misestimation of uncertainty[12]. Despite the potential benefits of using modelling results to improve the treatment and diagnosis of pathological anxiety, a major challenge remains due to the limited understanding of the neural processes underlying the computational alterations associated with anxiety.

Here, we build on recent progress in rhythm-based formulations of Bayesian predictive coding (PC) to identify sources of oscillatory modulations associated with altered learning in a volatile environment in subclinical trait anxiety. In a Bayesian PC framework, belief updates are informed by the discrepancy between predictions and outcomes—represented as prediction errors (PEs)—and weighted by precision (inverse variance or uncertainty of a belief distribution)[17–19]. The normative hierarchical updating policy of PC is thought to be orchestrated by distinct neural frequencies at particular cortical layers[19,20]. Evidence from human MEG/EEG and monkey local field potential (LFP) studies suggest that feedforward PE signals are encoded by faster gamma oscillations (>30 Hz), while backward descending predictions are expressed in lower alpha (8–12 Hz) and beta-band (13–30 Hz) oscillations[19,21–25]. Animal studies provide further evidence of this spectral dissociation, with alpha/beta activity in infragranular layers functionally inhibiting the processing of sensory input spiking, suppressing gamma oscillations in supragranular layers[26–29]. Precision is also encoded in alpha and beta oscillations[20,30]. As precision values weight the transmission of PEs[31], the composite precision-weighted PE (pwPE) signal may, as recent work suggests, be represented in antithetical modulation of gamma and alpha/beta power[23].

Crucially, although the oscillatory correlates of PC have been primarily investigated in the sensory domain, a similar mechanism in the medial prefrontal cortex (mPFC) has been shown to explain decision-making processes during exploration-exploitation[32]. In a reward-based learning task, we recently found that beta oscillations were atypically increased in state anxiety during the encoding of relevant pwPE signals[33,34]. In ref. [34]. there was also preliminary evidence for amplified beta activity maintaining (biased) predictions about the tendency of a stimulus-reward mapping in state anxiety. The role of gamma oscillations in mediating altered learning in anxiety through PE signalling remains, however, speculative. Due to the antithetic nature of gamma and alpha/beta activity in the human and non-human primate cortex[25,35–37], we predict that anxiety-related changes in alpha and beta activity during encoding pwPE should be accompanied by opposite effects in gamma. Moreover, given the relevance of precision weighting signals in explaining a manifold of psychiatric conditions[38–42], we expect that diminished or amplified precision weighting in anxiety during learning will be associated with changes in 8–30 Hz activity. This would result in biased predictions in this condition, possibly reflected in changes in alpha and beta oscillations.

The contribution of different brain regions to the frequency-domain expression of computational learning alterations in anxiety remains largely unknown. We hypothesise that neural sources that overlap with the neural circuitry of anxiety, decision making under uncertainty and reward-based learning, including the ventromedial, dorsomedial PFC (vmPFC, dmPFC), orbitofrontal cortex (OFC), and anterior cingulate cortex (ACC), will play a crucial role in the expression of altered oscillatory correlates of Bayesian PC during decision making in anxiety[1,32,43–48].

Here we test these hypotheses using computational modelling and source-level analysis of oscillatory responses in MEG. We investigated a low and high trait anxious group (LTA, HTA) on a binary probabilistic reward-based learning task under volatility. To assess whether trait anxiety interferes with reward-based learning performance through biased estimates of different forms of uncertainty, we modelled behavioural responses using a validated hierarchical Bayesian model, the Hierarchical Gaussian Filter (HGF[49,50]). This model was recently used to identify the sensor-level oscillatory correlates of Bayesian predictive coding in temporary anxiety states[34]. In the current work, we showed that HTA interferes with overall reward-based learning performance associated with more stochastic decisions and more pronounced lose-shift tendencies. These behavioural effects were explained by an overestimation of volatility and faster belief updating in HTA when compared to LTA.

We then extracted HGF estimates of unsigned pwPEs about stimulus outcomes, representing precision-weighted surprise about new information, and separately, the precision terms with which the PEs are weighted. These trajectories were used as input to a convolution model to estimate the time-frequency responses modulated by these computational learning quantities[51]. The convolution model was solved in the reconstructed source space using beamforming[52]. Our main finding is that HTA enhanced gamma responses in the ACC, dmPFC and lateral OFC during the encoding of unsigned pwPEs relative to LTA. The ACC additionally exhibited alpha/beta suppression during the encoding of pwPEs and precision weights in HTA. Our study thus identifies key brain regions expressing rhythm-based signatures of altered Bayesian PC during reward-based learning in anxiety.

## Results

**Initial learning adaptation in trait anxiety.** Thirty-nine participants (24 female, 15 male) completed a probabilistic binary reward-based learning task in a volatile learning setting[53–55] (reversal learning task), while we recorded their neural activity with MEG. Similarly to ref. [14], participants had to learn the probability that a blue or orange image in a given trial was rewarding (outcome win, 5 points reward; outcome lose, 0 points; complementary probabilities for both stimuli, P, 1-P; Fig. 1a). Participants expressed their choice by pressing the right or left button on a response box, corresponding with the position of the image they predicted to be rewarding on the current trial. The blue and orange stimuli were randomly presented to either the left or right of the screen. Participants were informed that the total sum of all their points would translate into a monetary reward at the end of the experiment. The task consisted of two task blocks with a total of 320 trials, 160 trials each. The stimulus-outcome contingency mapping changed four times across the 160 trials in each block (every 26-38 trials), and the five possible contingencies each block were 0.9/0.1, 0.1/0.9, 0.7/0.3, 0.3/0.7, and 0.5/0.5, as in refs. [14,54]. The order of contingency mappings within each block was generated pseudorandomly and separately for each participant (see example in Fig. 1b).

To assess our hypotheses that trait anxiety modulates belief updating during decision making in a volatile environment, we pre-screened the participants to form two experimental groups: low trait anxiety (LTA, which we defined as score below 36 in the

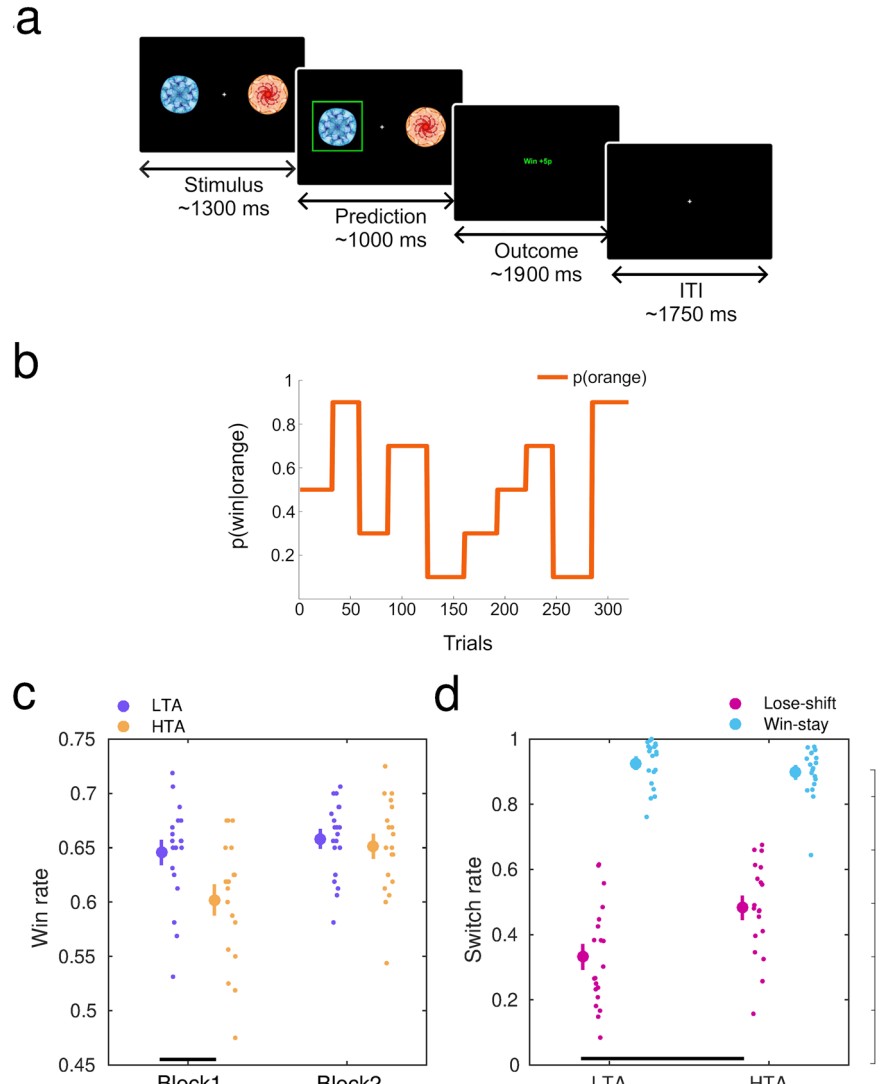

**Fig. 1 Trait anxiety modulates the win rate and the win-stay/lose-shift rates during reward-based learning. a** Behavioural task structure. The task was a standard reversal learning task. Participants were instructed to predict which of two images was the rewarding stimulus (win = 5 pence or 5p) on the current trial. The stimuli (blue or orange fractal) were randomly presented to either the left or right of the screen. They remained on the screen until a response was provided or the trial *timed out* (1300 ms ± 125 ms)—recorded as no-response. After they provided a left or right-side response, they immediately saw their chosen image highlighted in bright green, which remained on screen for 1000 ms (± 200 ms) before the outcome was displayed. The outcome, either win or lose, was shown in the middle of the screen for 1900 ms (± 100 ms) in green and red, respectively. Each trial ended with a fixation cross and an inter-trial interval (ITI) of 1750 ms (± 250 ms). **b** The probability governing the likelihood of the orange stimulus being rewarded, p(win|orange), is displayed in one example participant. The probability values p(win|orange) and p(blue|orange) in one trial were reciprocal: p(blue|orange) = 1 − p(win|orange). Probability mappings changed pseudorandomly every 26-38 trials and took the values 0.9/0.1, 0.7/0.3, 0.1/0.9, 0.3/0.7, and 0.5/0.5 in each block of 160 trials. See individual traces of contingency changes in Supplementary Fig. 1a. **c** High trait anxiety (HTA, yellow; $N = 19$ participants), and low trait anxiety (LTA, purple, $N = 20$ participants) modulated win rates differently as a function of the Block factor (significant interaction, $P = 0.0114$, 2 × 2 factorial analysis with synchronised rearrangements). There were also significant main effects of Block and Group. Post-hoc analyses demonstrated a significantly lower win rate in HTA relative to LTA in the first task block (Block 1, $P_{FDR} = 0.015 < 0.05$, permutation tests; denoted by the black bar at the bottom), but not in block 2. Within-group analyses further revealed that HTA participants significantly improved their win rate from block 1 to 2 ($P_{FDR} = 0.0036 < 0.05$, paired permutation test), whereas the win rate did not change significantly in LTA ($P_{FDR} > 0.05$). Data in each group are represented using the average (large dot) with SEM bars. To the right are individual data points to display dispersion. **d** Win-stay (blue) and lose-shift (magenta) rates in each anxiety group. The rates were estimated as the number of trials in that category relative to the total number of trials in the outcome type (e.g., lose-shift rate: number of lose-shift trials divided by the total number of lose events). High trait anxiety was associated with greater lose-shift rates relative to LTA ($P_{FDR} = 0.0034 < 0.05$, permutation tests), while win-stay rates did not significantly change as a function of anxiety ($P_{FDR} > 0.05$). Between-group differences are marked by the bottom (lose-shift) bars.

trait subscale of the Spielberger State Trait Anxiety Inventory[56], STAI, range 0–80; "Methods") and high trait anxiety (HTA, defined as a STAI trait score above 45; "Methods"). Both LTA and HTA samples were matched in age and the proportion of males and females (LTA, $N = 20$, 22.1 yrs [standard error of the mean

or SEM, 0.4 yrs], 12 female; HTA, $N = 19$, 21.7 [0.4] yrs, 12 female).

There were no systematic differences between groups in the order of contingency mappings (Supplementary Fig. 1). In 4/20 LTA and 3/19 HTA participants, however, the probabilistic

mapping did not change from block 1 to 2, and thus these participants encountered a total of eight contingency mapping changes across the 320 trials, while 16/20 LTA and 16/19 HTA individuals encountered nine probabilistic changes overall. Control analyses provided strong evidence in support of the null hypothesis that both groups were exposed to the same probabilistic mapping over time. There was also moderate evidence that both groups experienced on average an equal amount of true volatility (Supplementary Results: Validation analyses). Additional control analyses further supported that the main behavioural and computational group results were not confounded by individual differences in the pseudorandomised order of contingency mappings (Supplementary Results: Validation analyses). In addition, both samples did not differ during task completion in physiological changes in heart-rate variability (HRV and high-frequency HRV) previously associated with temporary states of anxiety[57] ("Methods", Supplementary Results: Measures of Anxiety, Supplementary Fig. 2).

Participants in each anxiety group exhibited different win rates (percentage of rewarded trials) depending on the task block (significant interaction effect of Block and Anxiety, $P = 0.0114$; non-parametric $2 \times 2$ factorial test with synchronised rearrangements[58], 5000 permutations). In addition, we observed a significant main effect of Block ($P = 0.0036$), and a significant Group effect ($P = 0.0280$, Fig. 1c). Follow-up post-hoc analysis with pair-wise permutation tests revealed a significantly smaller win rate in HTA during block 1 relative to LTA ($P = 0.015$, significant after control of the false discovery rate across multiple post-hoc tests, hereafter denoted by $P_{FDR} < 0.05$; non-parametric effect size estimator, $\Delta = 0.73$, CI = [0.64, 0.89]; "Methods"). By contrast, during the second block there was no significant between-group difference ($P_{FDR} > 0.05$, Fig. 1c). In addition, HTA individuals exhibited a pronounced increase of the win rate from block 1 to 2 ($P_{FDR} = 0.0036 < 0.05$, paired permutation test; paired-samples effect size $\Delta_{sup} = 0.74$, CI = [0.65, 0.87]), while this effect was not observed in LTA ($P_{FDR} > 0.05$). The individual and group average win rates were well below the ceiling win rate (mean 0.74 [SEM 0.001], maximum 0.76, measured from the true reward contingency settings). These results demonstrate that HTA exhibited poorer reward-based learning performance relative to LTA mainly due to differences in block 1, suggesting an initial adaptation deficit. HTA individuals, however, improved considerably during block 2 leading to higher win rates that failed to differ significantly from rates in LTA.

High win rates in a fast-changing environment could be associated with a tendency to express win-stay/lose-shift behaviour more[13,59,60]. To assess this, we calculated the win-stay and lose-shift rates, which were normalised separately for each outcome type: win or lose[61]. In HTA we found a significantly higher lose-shift rate when compared with LTA ($P_{FDR} = 0.0034 < 0.05$, $\Delta = 0.76$, CI = [0.58, 0.89], Fig. 1d), but no significant differences in the win-stay rate ($P_{FDR} > 0.05$). The higher lose-shift rate in HTA relative to LTA was strikingly similar across contingency phases (Supplementary Fig. 3). Thus, across the experiment, HTA individuals consistently switched more than LTA individuals after losing in a trial. This effect carried over to the total switch rate, which was significantly higher in HTA than LTA ($P_{FDR} = 0.0134 < 0.05$, $\Delta = 0.71$, CI = [0.55, 0.82]; mean switch rate in each group and SEM: 0.24 [0.02] in HTA, 0.16 [0.02] in LTA). This result was, however, mainly accounted for by group differences in lose-shift rates, as shown above. Post-hoc analyses demonstrated that the increased tendency to shift following lose outcomes in HTA relative to LTA, like for the general switch tendency, did not change throughout the experiment, despite HTA exhibiting an initial adaptation

deficit (expressed in lower win rates) that was overcome towards second block (Supplementary Results).

**Differential effects of trait anxiety on learning are best described by a hierarchical Bayesian model wherein decisions are driven by volatility estimates.** We next aimed to determine whether learning differences in our anxiety groups could be accounted for by changes in estimates of different forms of uncertainty. Overestimating uncertainty in the environment may lead to anxious avoidance responses and individuals missing out on invaluable safety signals and rewarding feedback[10,12,62]. Alternatively, higher levels of estimated environmental uncertainty may inflate the degree to which new outcomes update beliefs[63,64]. Learning can also be influenced by a different form of uncertainty, related to our imperfect knowledge about the true states in the environment (informational uncertainty[50]).

To assess different forms of uncertainty in our task, we modelled decision-making behaviour with the Hierarchical Gaussian Filter (HGF)[49,50]. This model allowed us to characterise individuals' trial-by-trial learning of the probabilistic stimulus-outcome mapping and its volatility. Volatility here represents the rate of change of the tendency towards a contingency mapping[14,50]. In the HGF, higher levels of volatility are associated with faster learning about the probabilistic relationships, whereas a stable environment would attenuate learning about the reward contingencies. The rationale for choosing the HGF as a hierarchical Bayesian modelling framework was based on its suitability to identify alterations in different types of uncertainty during decision-making behaviour in a very similar task in temporary anxiety states[14,34].

In the HGF, the individual trial-wise trajectories of the beliefs about the probabilistic mapping (HGF level $i = 2$) and log-volatility ($i = 3$) are represented by their sufficient statistics: $\mu_i$ (mean, commensurate to a participant's expectation) and $\sigma_i$ (variance, termed informational or estimation uncertainty for level 2; uncertainty about volatility for level 3; inverse of precision, Fig. 2b). The inverse variance is termed precision, $\pi_i$. Belief updating on each level $i$ ($i = 2$ and 3) and trial $k$ is driven by PEs modulated by precision ratios, weighting the influence of precision or uncertainty in the current level and the level below:

$$\triangle \mu_i^k = \mu_i^{(k)} - \mu_i^{(k-1)} \propto \frac{\hat{\pi}_{i-1}^{(k)}}{\pi_i^{(k)}} \delta_{i-1}^{(k)} \qquad (1)$$

Following Eq. (1), the expectation of the posterior mean on level $i$, $\mu_i^{(k-1)}$, is updated to its current level $\mu_i^{(k)}$ proportionally to the prediction error of the level below, $\delta_{i-1}^{(k)}$. The influence of PEs is weighted by the ratio of precision values, with the *prediction* (denoted by "^") of the precision of the level below in the numerator, and the precision of the current level (inverse uncertainty, $\sigma_i$) in the denominator. In the HGF for binary outcomes, the precision ratio updating beliefs on level 2 in Eq. (1) is reduced to $\sigma_2^{(k)}$. Accordingly, the posterior mean of the belief about the stimulus-reward contingencies is updated via PE about stimulus outcomes and scaled by the degree of informational uncertainty. For level 3, the precision ratio is proportional to the uncertainty about volatility, $\sigma_3^{(k)}$ (inverse precision on level 3: $1/\pi_3^{(k)}$). In the HGF update equations, the precision-weighted PE term updating level $i$ is typically labelled $\varepsilon_i$. We use this term hereafter.

The coupling function between levels 2 and 3 is as follows (dropping index $k$ for simplicity):

$$f_2(x_3) \overset{\text{def}}{=} \exp(\kappa x_3 + \omega_2) \qquad (2)$$

In Eq. (2), $\omega_2$ represents the invariant (tonic) portion of the log-volatility of $x_2$ and captures the size of each individual's stimulus-

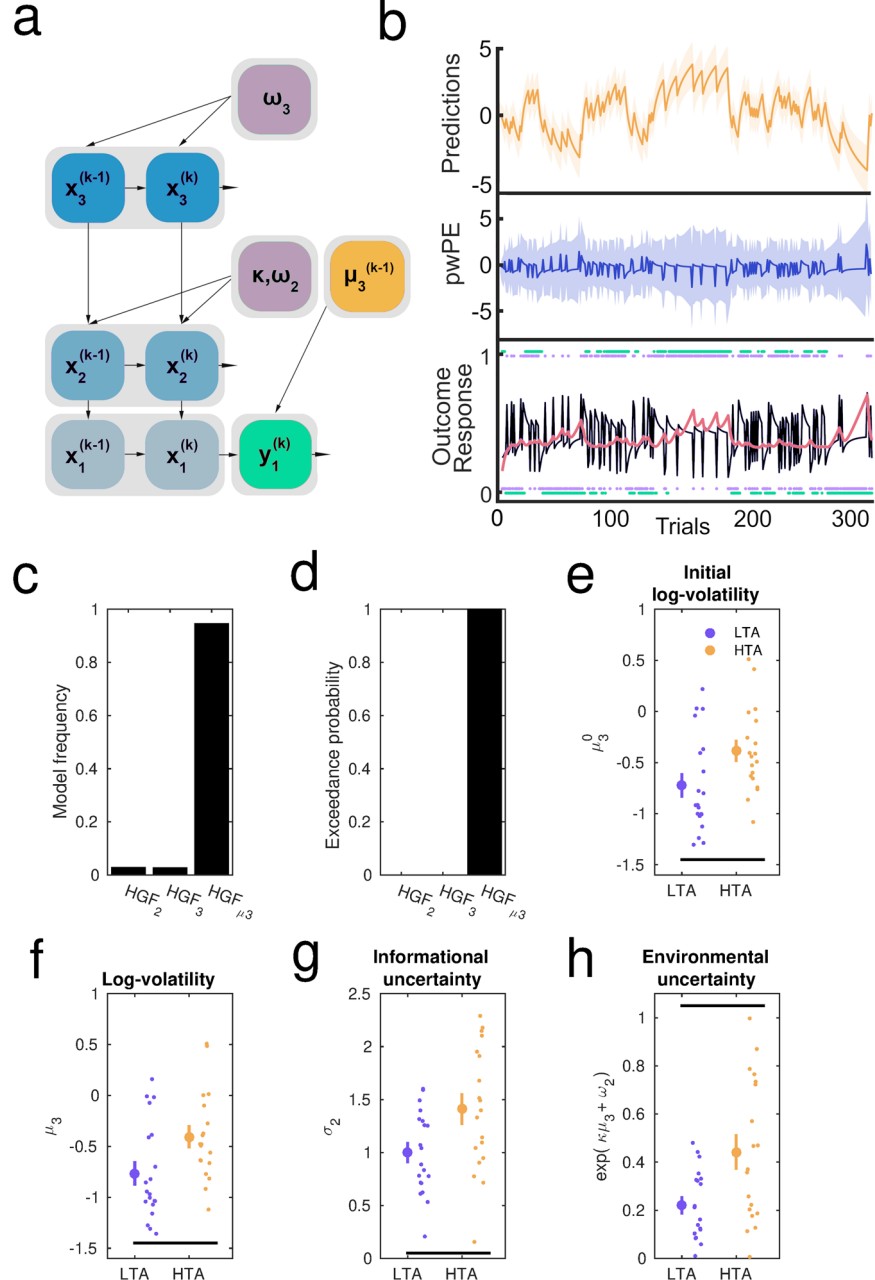

**Fig. 2 Hierarchical Gaussian Filter: Winning model and results. a** Representation of the three-level Hierarchical Gaussian Filter for binary outcomes with inverse decision noise being a function of the log-volatility prediction (estimate on the previous trial, $k$-1), $\mu_3^{(k-1)}$. **b** Associated trajectories of relevant HGF outputs the total 320 trials in a representative participant. At the lowest level, the inputs correspond to the rewarded outcome of each trial (1 = blue, 0 = orange; shown as purple dots). The participant's responses y are shown in green dots tracking those trial outcomes. The black line indicates the series of prediction errors (PE) about the stimulus outcome, and the salmon pink line the precision weight on level 2. The middle layer of (b) shows the trial-wise HGF estimate of pwPE about stimulus outcomes (pwPE updating level 2, simply termed pwPE in the graphic; deep blue). For our main GLM convolution analysis, we used unsigned values of pwPE updating level 2 as one of the parametric regressors. The precision ratio included in the pwPE term weights the influence of prediction errors about stimulus outcomes on the expectation of beliefs on level 2. Predictions about the tendency towards a stimulus-reward contingency on level 2($\hat{\mu}_2$) are displayed on the top level (yellow). We took the absolute values of this quantity as our main parametric regressor (labelled simply Predictions in the graphic) in a separate exploratory GLM analysis. **c, d** Bayesian model selection (BMS). The panels show the model frequency (c) and exceedance probability (d) for each of the HGF models we tested: the 2-level HGF (HGF$_2$), the 3-level HGF (HGF$_3$), and the 3-level HGF informed by trial-wise estimates of volatility (HGF$_{\mu 3}$) given in black. The HGF$_{\mu 3}$ model best explained the data (exceedance probability = 1; expected frequency = 0.95). **e** HTA individuals (yellow; $N$ = 19 participants) had a greater initial expectation or prior on log-volatility than LTA (purple, $N$ = 20 participants; $P_{FDR}$ = 0.024 < 0.05; group effects denoted by the black line at the bottom). Data in each group are represented by the mean (large dot) with SEM bars. Individual data points are illustrated to the right. **f** Over time, the posterior mean on log-volatility ($\mu_3$) in HTA remained significantly higher relative to LTA ($P_{FDR}$ = 0.019 < 0.05). **g** Informational (estimation) belief uncertainty about the stimulus outcome tendency was greater in HTA compared with LTA ($P_{FDR}$ = 0.0138 < 0.05). **h** The HTA individuals were also significantly more uncertain about the environment ($P_{FDR}$ = 0.0052 < 0.05). No significant differences were found in uncertainty about volatility ($\sigma_3$) or the tonic learning rates at levels 2 ($\omega_2$) and 3 ($\omega_3$).

outcome belief update independent of $x_3$. The $\kappa$ (*Kappa*) parameter establishes the strength of the coupling between $x_2$ and $x_3$, and thus the degree to which estimated environmental volatility impacts the learning rate about the stimulus-outcome probabilities—here $\kappa$ was fixed to one as in previous work[14,54]. On level 3, the step size of $x_3$ depends on the exponential of a positive constant parameter $\omega_3$ (the lower $\omega_3$ the slower participants update their beliefs about volatility). Further details are provided in "Methods".

To describe how participants updated their beliefs about the reward contingencies, we first used two types of Bayesian perceptual models: the 3-level HGF and a reduced 2-level HGF with fixed volatility[14] ("Methods"). Next, to explain the trial-by-trial response data in our participants, we combined the 3-level HGF with two alternative forms of response model (one alternative response model for the 2-level HGF), describing different ways in which participants' beliefs are mapped to decisions. The mapping was governed by a unit-square sigmoid function: (i) with a fixed parameter $\zeta$ that can be interpreted as inverse decision noise that shapes choice probability[49,50] ("Methods"); (ii) where the inverse decision noise is a function of the prediction of log-volatility[65]: $e^{-\mu_3^{(k-1)}}$, thus depending on the participant's trial-wise beliefs on volatility—termed HGF$_{\mu3}$. Response model (i) is useful because it captures how deterministically a response is associated with the predictive probability of the next outcome. Individuals with higher $\zeta$ values are more likely to choose the response in agreement with their outcome prediction on the current trial. On the other hand, participants could exhibit changes over time in how they map beliefs to choices, and these changes could be a function of the estimated level of volatility, $\mu_3$. For instance, when individuals estimate the environment to be more stable, their responses could follow more deterministically the outcome predictions for the current trial. Yet if their volatility estimate increases over the course of the session, their mapping could be more stochastic. This scenario was captured by response model (ii), introduced in ref. [65]. This resulted in three types of Bayesian perceptual+response models (3-level HGF with [i] and [ii], termed HGF$_{\mu3}$; 2 level HGF with [i]). Details on the fixed and estimated model parameters are provided in "Methods", and the prior settings are listed in Table S1.

While previous HGF studies[23,54,55], including our own work[14], also considered widely used and relatively simple reinforcement learning models, the model comparison approaches consistently demonstrated that the HGF models described the data best. Accordingly, we limited our model space to three HGF models. In future work, it would be important to assess the performance of alternative Bayesian models that were designed—as the HGF—to characterise learning in volatile environments. One such model is the one proposed by Piray and Daw to jointly estimate volatility and stochasticity[66]. A direct comparison between these different Bayesian models is not straightforward at this point as model inversion for the HGF uses variational Bayes, while the probabilistic model by Piray and Daw uses Monte Carlo sampling to estimate belief distributions[66]. Reformulating models to the same Bayesian inference framework to allow for model comparison is challenging[67], and not feasible in the current study.

As in previous work with the HGF[54,55,65], we evaluated the model space using random effects Bayesian model selection[68,69] (BMS, "Methods"). This approach uses the log-model evidence obtained for each participant and model to obtain two quantities: (i) Exceedance probability, as the probability that one model explains the data better than other models; (ii) Expected frequency, i.e., conditional estimate of how frequently one model wins against the other models. It is standard in HGF papers to

obtain both quantities and choose the model that outperforms other models in both parameters. Here, the model that was more likely to explain the behavioural data among participants was the 3-level HGF coupled with a response model where decisions are informed by trial-wise estimates of volatility[65] (HGF$_{\mu3}$; BMS results: exceedance probability = 1; expected frequency = 0.95; Fig. 2c, d; similar results were observed when assessing BMS in each group separately; Supplementary Results).

In the winning HGF$_{\mu3}$ response model a greater expectation on log-volatility for the current trial is associated with higher decision noise (lower inverse decision noise parameter), leading to a noisier mapping between beliefs and responses. On the other hand, when a participant has a lower expectation on volatility governing the stimulus-reward contingencies, she will exhibit a more deterministic coupling between her current belief and subsequent response[65]. In the context of trait anxiety, the BMS result demonstrates that inferring the underlying environmental statistics and deciding upon responses is best described by a hierarchical model in which the mapping from beliefs to responses is a function of the prediction of volatility.

**Overestimation of environmental volatility in high trait anxiety.** HTA individuals had a greater initial estimate on volatility (free parameter $\mu_3^{(0)}$) than LTA participants ($P_{FDR} = 0.024 < 0.05$, $\Delta = 0.72$, CI = [0.54, 0.87]; Fig. 2e). Over trials, we observed that the posterior mean on log-volatility estimates, $\mu_3$, remained higher in the HTA group relative to the LTA group ($P_{FDR} = 0.019 < 0.05$, $\Delta = 0.74$, CI = [0.55, 0.87]; Fig. 2f). No between-group difference was found in the associated third-level model parameter $\omega_3$ ($P > 0.05$). In the HGF$_{\mu3}$, an estimated greater level of task environmental change HTA relative to the LTA group suggests that choice probability in HTA individuals is more stochastic. In other words, compared to LTA, HTA participants chose more often responses that were less likely to be rewarded based on their predictions for the trial.

The increased response stochasticity in HTA converges with our findings on lose-shift rates, which demonstrated an overall higher tendency to switch in HTA following lose trials—even if this goes against the current belief on the tendency of the stimulus-reward contingency. It is also aligned with the related finding of a higher overall switch rate (change independently of the outcome) in HTA individuals. As a post-hoc analysis, we conducted a non-parametric correlation across all participants between the overall switch rate and the average estimate of log-volatility, $\mu_3$. We found a significant association between both variables, as expected (non-parametric Spearman rank correlation $\rho = 0.89$, $P < 0.00001$; $N = 39$). Our behavioural findings thus concur with the modelling results showing that HTA individuals exhibit an overestimation of volatility, which in the HGF$_{\mu3}$ leads to more 'stochastic' switching responses. This outcome is mainly driven by switching following a lose outcome.

**Misestimation of different types of uncertainty in trait anxiety can promote learning despite an initial adaptation deficit.** Informational uncertainty about the stimulus-outcome contingency, $\sigma_2$, drives the pwPEs updating level 2, with larger $\sigma_2$ values contributing to greater update steps. Participants with HTA overestimated informational uncertainty relative to LTA individuals ($P_{FDR} = 0.0138 < 0.05$, $\Delta = 0.72$, CI = [0.55, 0.86], Fig. 2g). This result suggests that new information has a greater impact on the update of beliefs about the tendency towards a stimulus-reward contingency (level 2), promoting faster learning on that level.

An additional important type of uncertainty governing learning in our task is uncertainty about the task environment,

termed environmental uncertainty[50]: $\exp(\kappa\mu_3^{(k-1)} + \omega_2)$. Here $\mu_3^{(k-1)}$ denotes the mean estimate on log-volatility in the previous trial, $k-1$, which is the mean expectation for trial $k$. This type of uncertainty is also a function of the tonic volatility, $\omega_2$. We found that the HTA group had greater environmental uncertainty when compared with LTA participants ($P_{\text{FDR}} = 0.0052 < 0.05$, $\Delta = 0.74$, CI = [0.55, 0.88], Fig. 2h). There was, however, no significant between-group difference in the related parameter $\omega_2$, or in uncertainty about the volatility estimate, $\sigma_3$ ($P > 0.05$ in both cases). The latter outcome suggests that trait anxiety had no significant effect on the speed of updates about volatility ($\sigma_3$ weights prediction errors updating level 3). Rather, trait anxiety led individuals to overestimate the level of volatility in the environment already from the start, and this estimate remained high throughout the task.

**Source analysis results**. Having established that HTA is associated with a relative faster update of beliefs about the tendency of the stimulus-outcome contingency through enhanced informational uncertainty and more stochastic behaviour due to higher expectation on volatility, we next aimed to identify the source-level neural oscillatory processes accompanying these computational effects. Accordingly, we assessed the source-reconstructed neural oscillatory representations of pwPEs and precision weights during reward-based learning in our anxiety groups. Similarly to ref. [34], this was achieved using linear convolution models for oscillatory responses[51]. This approach is an adaptation of the classical general linear model (GLM) used in fMRI analysis to time-frequency (TF) data and has been successfully used in previous EEG and MEG research[23,70]. It allows assessing the modulation of TF responses on a trial-by-trial basis by one specific explanatory regressor while controlling for the effect of the remaining regressors included in the model ("Methods").

To relate precision terms and pwPEs to oscillatory neural activity, as well as to explore the effect of predictions, we selected the individual HGF trajectories of the relevant parameters as parametric regressors in a GLM. According to Eq. (1), the weights on the PEs are updated proportionally to the uncertainty $\sigma_i$ on each level, $i = 2, 3$. In other words, greater uncertainty (lower precision) about beliefs on level 2 or 3 enhances the impact that PEs have on updating that level. We therefore included informational uncertainty, $\sigma_2$, and uncertainty on level 3, $\sigma_3$, as the relevant (inverse) precision terms for the GLM. We also chose the unsigned pwPE on level 2 (termed $|\varepsilon_2|$), following previous work[23,71], while the pwPE regressor on level 3 was excluded due to multicollinearity[34,72] ("Methods"). To additionally explore the effect of predictions, we selected the unsigned predictions on level 2 ($|\hat{\mu}_2|$) about the tendency towards a certain stimulus-reward contingency (henceforth: 'predictions'; "Methods"). The absolute correlation values between each pair of chosen regressors was below 0.2, which allowed us to include them as independent predictors in the GLM.

Based on prior work[23,34], we hypothesised that the neural responses correlated with pwPEs and precision terms would be observed in a time interval following the outcome presentation, whereas the effect of predictions before observing the outcome could be determined by analysing the post-stimulus (pre-outcome) interval[23,34,42]. A scheme of the hypothesised timeline of effects is presented in Supplementary Fig. 4.

A GLM on the continuous time series could include these parametric regressors, along with discrete regressors representing behavioural events at their respective time onsets: stimulus cues, responses, and outcome cues. However, given that convolution modelling for oscillatory responses is computationally expensive and we hypothesised dissociable temporal effects of pwPEs and

predictions—which was observed in refs. [23,34]—we opted to run two separate GLMs in different non-overlapping time windows: an outcome-locked and a stimulus-locked GLM.

The main outcome-locked GLM evaluated the effect of parametric regressors $|\varepsilon_2|$, $\sigma_2$, and $\sigma_3$, on the TF responses, while it controlled for the effect of discrete outcome events (win, lose, no response). Next, in an exploratory analysis, we implemented a stimulus-locked GLM to assess the neural oscillatory processes correlated with the parametric regressor $|\hat{\mu}_2|$. This model additionally included discrete regressors denoting the stimuli presentation (blue image on the left or right side), the participant's response (left, right, no response), and outcome cues (win, lose, no response) at their respective onset ("Methods").

The GLM analyses were conducted in the source space after applying linearly constrained minimum norm variance (LCMV) beamformers[52] to the time series of concatenated epochs of MEG data ("Methods"). To reduce the data dimensionality, the convolution models were estimated in a set of brain regions previously associated with anxiety, decision making and reward processing: (1) ACC, (2) OFC and related vmPFC, and (3) dmPFC. The ACC and medial PFC have been consistently shown to be involved in pathological and adaptive/induced anxiety, but also in emotional and reward processing and decision making[1,43,46,73]. Within the medial PFC, the vmPFC represents reward probability, as well as magnitude, and outcome expectations[32,47]. The dmPFC, on the other hand, has been shown to elicit gamma activity that correlates with unsigned reward prediction errors during exploration-exploitation[32]. The OFC is also particularly relevant in our study, as it has been associated with emotional processing, reward and punishment processing[48]. In particular, the medial OFC (mOFC) encodes reward value, whereas the lateral OFC (lOFC) encodes nonreward and punishment[48,74]. The vmPFC and OFC are also considered to play a central role in the "uncertainty and anticipation model of anxiety" (ref. [1]). These regions of interest (ROIs) corresponded to five bilateral labels (10 in total) in the neuroanatomical Desikan-Killiany–Tourville atlas[75] (DKT), which we chose to parcellate each participant's cerebral cortex using the individual T1-weighted MRI (Fig. 3a, b; "Methods").

We tested the hypothesis that high levels of trait anxiety are associated with changes in gamma and concomitant alpha/beta activity during encoding pwPE signals. In addition, we hypothesised that trait anxiety modulates alpha/beta oscillatory activity during the representation of precision weights.

**Unsigned precision-weighted prediction errors about stimulus outcomes**. A between-subject independent sample cluster-based permutation test between 8–100 Hz on the TF responses to $|\varepsilon_2|$ revealed a significant decrease at 10–16 Hz in the HTA group relative to LTA in the caudal portion of the ACC (cACC, two negative spectral-temporal clusters, $P = 0.01$ and 0.008, two-sided test, FWER-controlled, 3D data: 10 labels × samples × frequency bins). The latency of the significant effect was 450–550 ms and 1200–1400 ms *post-outcome* (Fig. 3c, d). A second significant effect in the low-frequency range was found in the lateral OFC, due to relative increased 10–22 Hz activity in HTA (positive cluster within 1450–1700 s, $P = 0.008$, two-sided test, FWER-controlled; Fig. 3e, f). Crucially, the latency of these effects extended for at least two full cycles of the central cluster frequency. In the gamma range, we observed prominent increases in TF responses in HTA as compared to LTA participants in the cACC, lOFG and dmPFC (positive clusters, $P = 0.001$, 0.005, and 0.001, two-sided test, FWER-controlled; Fig. 3c–h; the dmPFC is represented by the anatomical label 'superior frontal gyrus', SFG;

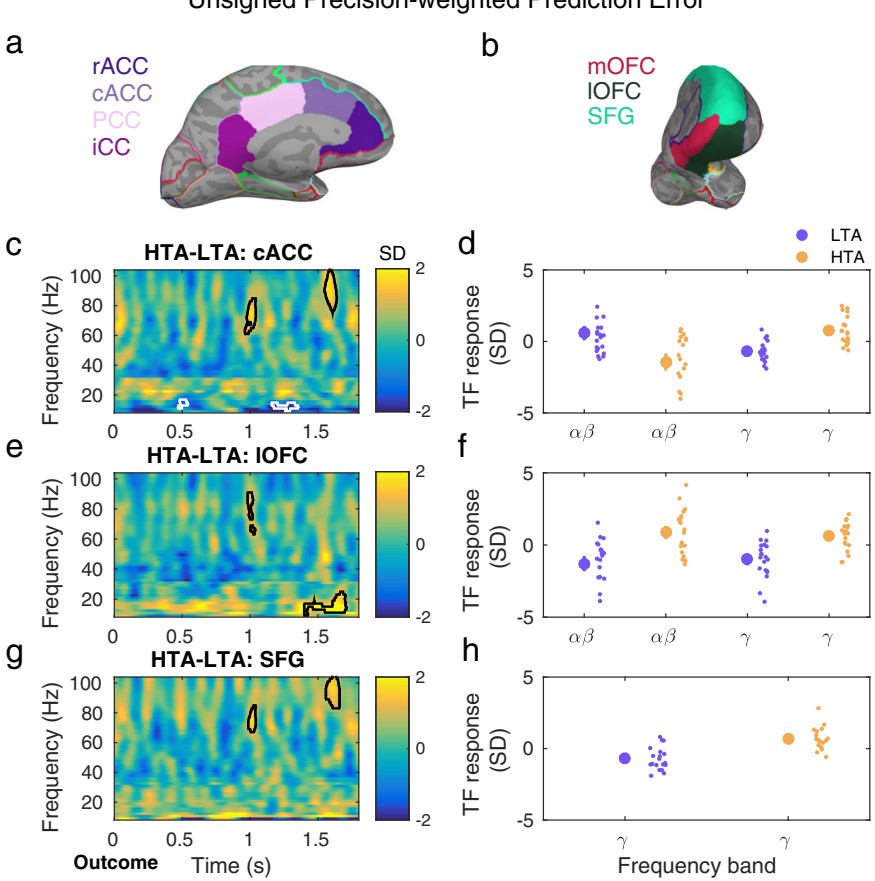

**Fig. 3 Gamma activity is modulated by unsigned precision-weighted prediction errors about stimulus outcomes and is enhanced with high trait anxiety. a, b** Source reconstruction of MEG signals was carried out with linearly constrained minimum norm variance (LCMV) beamforming. The statistical analysis of the convolution GLM results focused on brain regions that overlap with the circuitry of anxiety and decision making under uncertainty: ACC, OFC (lateral and medial portions: lOFC, mOFC), SFG. Panels a and b illustrate the corresponding anatomical labels in the neuroanatomical Desikan-Killiany–Tourville atlas (DKT), which we chose to parcellate each participant's cerebral cortex using the individual T1-weighted MRI. Panels **c, e, f** display between-group differences in the time-frequency (TF) images that summarise the individual oscillatory responses to the unsigned precision-weighted PEs about stimulus outcomes. TF images are shown in the 8–100 Hz range, including alpha (8–12 Hz), beta (14–30 Hz) and gamma (32–100 Hz) activity. TF images were normalised with the mean and standard deviation (SD) of the activity in a [−300, −50] ms pre-outcome interval, and thus are given in SD units. Convolution modelling and TF transformation were conducted in the range 8–120 Hz in frequency bands of 2 Hz, following LCMV beamforming. **c** In the cACC, high relative to low trait anxiety was associated with greater gamma responses at ~1 and 1.6 s during outcome feedback processing (cluster-based permutation testing, two clusters, $P = 0.001$, FWER-controlled; $N = 19$ HTA and 20 LTA independent samples). Significant between-group effects are denoted by the black and white contour lines in the TF images. The gamma-band effects were accompanied by a decrease in alpha-beta activity (10–16 Hz) in HTA as compared to LTA individuals, and at ~0.5 and 1.3 s (cluster-based permutation testing, $P = 0.01$ and 0.008, FWER-controlled). **d** The relative gamma increase in HTA shown in c) was due to more positive gamma activity during encoding unsigned pwPE in HTA than in LTA. On the other hand, HTA individuals exhibited a negative change in 10–16 Hz responses, in contrast to the positive alpha and beta activity observed in LTA individuals. This resulted in the negative between-group effect in 10–16 Hz activity in (**c**). The large circles represent the mean (and SEM) TF response in the significant spectrotemporal clusters in c), shown separately for low frequency (alpha, beta) and gamma activity, and for each group (LTA: purple; HTA: yellow). Individual dots represent individual participant average values. **e, f** Same as **c, d** but in the lOFC. Significant clusters in 10–22 Hz and gamma ($P = 0.008$ and 0.005, respectively, FWER-controlled). **g, h** Same as **c, d** but in the SFG. Significant clusters in gamma frequencies ($P = 0.001$, FWER-controlled). rACC, rostral anterior cingulate cortex; cACC, caudal ACC; PCC, posterior CC; iCC, isthmus of the CC; SFG, superior frontal gyrus; lOFC, lateral orbitofrontal cortex; HTA, high trait anxiety; LTA, low trait anxiety.

"Methods"). The enhanced gamma modulation in HTA relative to LTA had a similar latency across these regions: it emerged at around 1000 ms within 60–80 Hz and at 1600 ms within 80–100 Hz. The gamma effects extended for at least 5 cycles at the central cluster frequency. No other effects were found. Importantly, a control analysis demonstrated that a different choice of the Fourier basis set to increase the temporal resolution on the GLM analysis of high-frequency gamma modulations revealed very similar results ("Methods"; Supplementary Fig. 5). Moreover, including the pwPE regressor on level 3 $\varepsilon_3$ instead of our choice of $|\varepsilon_2|$ also demonstrated similar results (Supplementary Fig. 6), as

expected, given the high correlations between both regressors ("Methods").

Next, we reasoned that the greater gamma activity observed in HTA in the cACC, dmPFC (SFG) and lOFC during encoding $|\varepsilon_2|$ could reflect an association between larger $|\varepsilon_2|$ values and a greater likelihood of switching responses in HTA. In the ACC, reward and value estimates guide choices, with higher ACC activity observed in trials leading to choices[76]. In addition, activity in the dmPFC represents value difference signals modulating motor responses[77]. We therefore asked whether trials leading to a response shift had larger $|\varepsilon_2|$ values, due faster belief

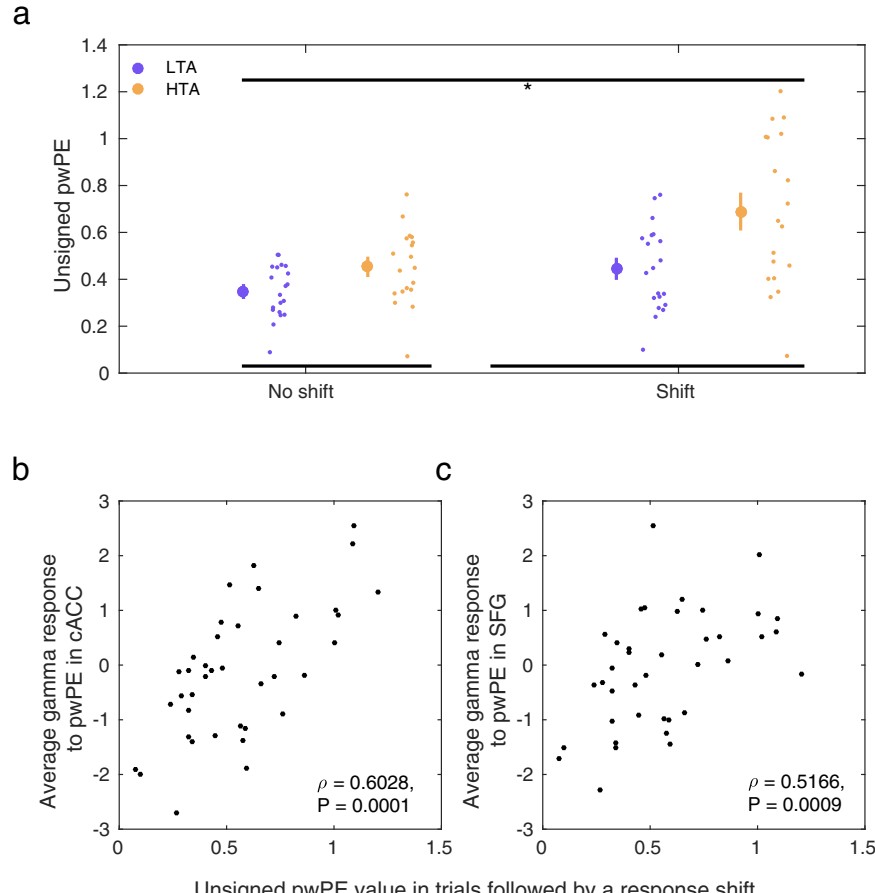

**Fig. 4 Trials leading to a shift in response choice are associated with larger unsigned pwPE values. a** The values of unsigned pwPE updating beliefs about stimulus-reward contingencies (denoted by $|\varepsilon_2|$) were larger in trials that were followed by a response shift (significant main effect of factor Shift, $2 \times 2$ synchronised rearrangements, $P = 0.0012$; denoted by the black line on top and the asterisk). High relative to low trait anxiety individuals also had larger $|\varepsilon_2|$ overall (main effect of Group, $P = 0.0068$; denoted by the black lines at the bottom) and a more pronounced dissociation between the $|\varepsilon_2|$ values in trials followed by a response shift or repetition (significant interaction, $P = 0.0200$). Data in each group are represented using the average (large dot) with SEM bars. To the right are individual data points to display dispersion. HTA, high trait anxiety (yellow; $N = 19$ participants); LTA, low trait anxiety (purple; $N = 20$ participants). **b** Non-parametric correlation between the average gamma response to the unsigned pwPE regressor in the cACC and the $|\varepsilon_2|$ values in trials followed by a response shift (Spearman $\rho = 0.6028$, $P = 0.0001$; the gamma average was estimated in the significant cluster of between-group differences, Fig. 3). **c** Same as (**b**) but for the dmPFC ($\rho = 0.5166$, $P = 0.0009$).

updating; we also assessed whether this effect was modulated by the Group factor. A $2 \times 2$ Group x Shift (trial followed by a response shift / no shift) analysis of unsigned pwPE values on level 2 demonstrated a significant main effect of the Group and Shift factors ($P = 0.0068$, $0.0012$ respectively; Fig. 4). A significant interaction effect was also observed ($P = 0.0200$). These results demonstrate that $|\varepsilon_2|$ was larger in trials followed by a shift in the choice made by participants; $|\varepsilon_2|$ was also greater in HTA participants overall. Moreover, the modulation of $|\varepsilon_2|$ values by the Shift factor was more pronounced in HTA. Complementing these results, we observed that individuals with a greater gamma modulation by $|\varepsilon_2|$ in the cACC and dmPFC had larger $|\varepsilon_2|$ values (non-parametric Spearman correlation: $\rho = 0.6028$, $P = 0.0001$ in the cACC; $\rho = 0.5166$, $P = 0.0009$ in the dmPFC; Fig. 4). Gamma responses in the lOFC were not associated with $|\varepsilon_2|$ values ($\rho = 0.21$, $P = 0.1396$).

**Modulation of informational uncertainty and uncertainty about volatility by anxiety.** The precision weight term scaling the influence that PEs have on updating beliefs on level 2 corresponds to the informational uncertainty estimate, $\sigma_2$. This regressor was correlated with an attenuation of low frequency activity in the

cACC in high relative to low trait anxiety individuals (two negative clusters, $P = 0.01$, two-sided test, FWER-controlled Fig. 5a). The between-group effect was observed at 14–16 Hz and with a latency of 1050–1170 ms, corresponding to the latency of the gamma effect of $|\varepsilon_2|$ (Fig. 3) and extending for 1–2 oscillation cycles. An additional relative TF supresion was found at 8–10 Hz around ~1.6 s. In both clusters, LTA participants had predominantly a positive alpha/beta activity response to the precision-weight regressor, whereas HTA individuals exhibited mainly an attenuation of this response (Fig. 5b).

By contrast, a relative HTA minus LTA increase in 10–16 Hz and 12 Hz activity was observed for $\sigma_2$ in the lateral OFC in both hemispheres (at ~0.4 and 1.5 s; $P = 0.02$ in each case, two-sided test, FWER-controlled; Fig. 5c, d). Exploratory analyses in anatomical labels outside of our ROIs showed a between-group effect of precision weights on level 2, $\sigma_2$, in alpha/beta activity exclusively in the posterior cingulate cortex (negative cluster, $P = 0.02$, two-sided test, *uncorrected*).

Uncertainty about volatility, $\sigma_3$, which weights the updates of beliefs on level 3, was associated with a significant between-group statistical effect in the dmPFC at 12–20 Hz with a latency of 0.27–0.5 and later at 1.55–1.8 s ($P = 0.007$ and $P = 0.004$ in each

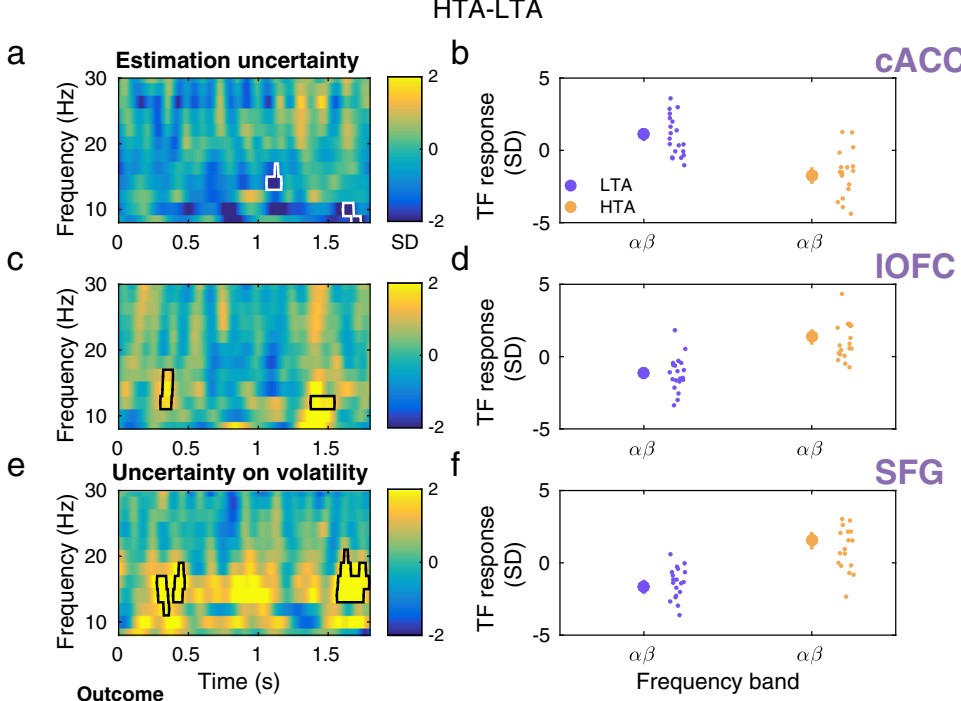

**Fig. 5 Modulation of alpha and beta activity during the representation of uncertainty in high trait anxiety. a** To investigate the effect of trait anxiety on the neural oscillatory correlates of precision weights modulating the influence that PEs have on updating predictions on each level, we included $\sigma_2$ and $\sigma_3$ as additional parametric regressors in our main GLM. Informational uncertainty, $\sigma_2$, is the precision ratio updating beliefs on level 2. For level 3, the precision ratio is proportional to the uncertainty about volatility, $\sigma_3$ (inverse precision on level 3: $1/\pi_3$; Eq. 1) The model also included continuous regressor $|\varepsilon_2|$ and discrete regressors coding for outcome events (win, lose, no response). TF images were normalised with the mean and standard deviation (SD) of the activity from −300 to −50 ms before the outcome feedback, and thus are given in SD units. In the caudal ACC (cACC), the representation of precision weights $\sigma_2$ was associated with a relative reduction of higher alpha and low beta activity in high trait anxiety, at latencies that were aligned with the gamma effects in Fig. 3 (~1.1 and 1.6 s; $P = 0.01$, FWER-controlled). **b** Average activity in the significant spectrotemporal clusters in a) separately for each group (LTA: purple, $N = 20$ participants; HTA: yellow, $N = 19$ participants). The large dot denotes mean and SEM as error bars. Individual dots represent individual participant average values. **c** In the lateral OFC (lOFC), we observed an increase in 12–16 Hz activity in HTA relative to LTA at 0.3–0.4 and 1.35–1.55 s ($P = 0.02$, FWER-controlled). **d** Average activity in the significant spectrotemporal clusters in (**c**) separately for each group. **e** Uncertainty about volatility, $\sigma_3$, which weights the updates of beliefs on level 3, was associated with a significant between-group statistical effect in the SFG at similar latencies than the $\sigma_2$ effects observed the lateral OFC ($P = 0.007$ and $P = 0.004$ for early and later cluster effects, respectively; FWER-controlled). HTA individuals exhibited increased beta activity to the $\sigma_3$ regressor relative to LTA participants. **f** The individual and group beta activity average in the significant clusters in (**e**) is displayed for each group separately.

case, two-sided test, FWER-controlled; Fig. 5e, f). The effect demonstrated that greater beta activity emerged in HTA relative to LTA participants in this brain region during encoding of uncertainty about volatility.

Regarding the discrete win and lose regressors in this main GLM, the TF images revealed between-group beta and gamma effects in brain regions that overlapped with those associated with the pwPE regressor, in line with predictive coding proposals[18], albeit with some polarity changes (Supplementary Fig. 7). Last, in an exploratory analysis, we assessed the modulation of theta (4–7 Hz) activity by the pwPE, as theta activity can facilitate encoding of unpredictable stimuli (akin to PE), driving gamma activity[25]. Convolution modelling revealed a general increase in the amplitude of theta activity to the unsigned pwPE regressor in HTA when compared to LTA (Supplementary Fig. 8; $P = 0.034$, uncorrected, rostral ACC and isthmus CC; note the reverse polarity effect for the Lose regressor; $P = 0.041$, uncorrected: Supplementary Fig. 9).

**Stimulus-locked predictions about reward tendency.** In a separate exploratory analysis, we tested the hypothesis that anxiety modulates alpha/beta oscillatory activity during the maintainance of predictions about the tendency of the stimulus-

outcome contingency[14], $\hat{\mu}_2$. In this stimulus-locked GLM, we observed a significant between-group difference in the beta-band TF responses to $|\hat{\mu}_2|$, due to greater beta activity in participants with high relative to low trait anxiety (Supplementary Fig. 10a, b; significant positive clusters at 100–200 ms and 600–680 ms post-stimulus, $P = 0.005$, FWER-controlled). This effect was limited to the cACC (right and left hemisphere) and contrasted with the pronounced drop in beta activity observed in HTA relative to LTA with the discrete stimulus regressor (Supplementary Fig. 10c, d; several significant clusters from 100 to ~700 ms post-stimulus; $P = 0.001$, FWER-controlled; These effects emerged before the feedback presentation at around ~1550 ms on average across participants). All significant between-group effects extended for at least one cycle at the relevant cluster frequency.

Last, the discrete response regressors (left, right) induced in each group a prominent alpha reduction prior to and during the button press, and a classic subsequent beta rebound (Supplementary Fig. 10e, f). There were no significant between-group differences in the TF images of these regressors. Neither in our ROIs (Supplementary Fig. 11; $P > 0.05$), nor in the additional DKT anatomical labels ($P > 0.05$).

## Discussion

Our study revealed that key brain regions of the anxiety and decision-making circuitry exhibit changes in oscillatory activity that can account for behavioural and computational effects of anxiety within a Bayesian predictive coding framework. We showed that HTA interferes with overall reward-based learning performance, which was associated with biased estimates of different forms of uncertainty. Inflated estimates of environmental volatility drove these changes, in line with previous reports that anxious learners overestimate volatility in all environments[78]. Noisier decisions and more pronounced lose-shift tendencies accompanied higher volatility estimates for HTA participants.

Recent proposals conceptualise some of the psychiatric symptoms in affective disorders as divergent hierarchical Bayesian inference, described by difficulties estimating uncertainty and balancing the influence of sensory input on belief updating[12,79]. These proposals extend to subclinical anxiety, given the considerable overlap of behavioural and neural effects in pathological and subclinical populations[1,46,73]. Our results are in line with these predictions, demonstrating a greater degree of informational uncertainty, $\sigma_2$, in HTA. HTA participants also overestimated environmental uncertainty and environmental volatility, $\mu_3$, already from the start ($\mu_3^{(0)}$). Greater informational uncertainty (smaller precision) drives faster update steps on the beliefs on the tendency of the stimulus-reward contingency[50]. Larger $\mu_3$ values also influence lower-level pwPEs, inflating the degree to which new outcomes update beliefs[63,64]. Thus, our results associate subclinical trait anxiety with faster updating of beliefs about stimulus-reward contingencies through an overestimation of informational uncertainty and environmental volatility.

Our trait anxiety results converge with findings from ref. [13], who described in clinical anxiety an inflexible adjustment of learning rates—remaining suboptimaly large—to volatility, as well as an inflated lose-shift rate. Induced anxiety states, by contrast, attenuate belief updating about the reward contingencies governing the environment[14]. In ref. [14], state-anxious individuals underestimated informational and environmental uncertainty. Similar to the state anxiety results, the somatic (physiological) component of trait anxiety has been linked to the underestimation of uncertainty and relative uncertainty between choices during exploration[15,16].

Here, biased estimates of uncertainty in HTA were associated with suboptimal switching behaviour, such as a pronounced lose-shift tendency. A hierarchical Bayesian model in which the mapping from beliefs to responses was a function of volatility best described the participants' behaviour in our task. As HTA individuals had a consistently larger prediction of volatility throughout the task, this model implied that, compared to LTA, the HTA group chose more often responses that were less likely to be rewarded based on their predictions for the trial. Increased response stochasticity in HTA agrees with its larger lose-shift rate and overall switch rate. This may explain the initially poorer task performance of this group, as higher levels of response switching combined with a high learning rate would make it difficult to infer the true probabilistic contingencies. In this scenario, distinguishing between meaningful environmental changes and outcome randomness would be more challenging. By modelling unpredictability, volatility[66], and subjective uncertainty (confidence ratings[16]) separately, follow-up work could determine whether a subjective misattribution of the causes of loss outcomes[80] could account for the increased choice stochasticity in anxiety.

By applying convolution models to explain amplitude modulations in time-frequency MEG responses[51,70], we were able to determine the effect of trait anxiety on the source-reconstructed neural oscillatory correlates of unsigned pwPE, informational uncertainty and uncertainty about volatility—while controlling for the simultaneous effect of discrete behavioural regressors[23,32,51]. Our analysis identified the cACC, dmPFC, and lOFC as brain regions accounting for computational alterations in reward-based learning in anxiety through changes in oscillatory activity. The results extend time-domain EEG and fMRI studies of Bayesian inference and predictive coding[55,65,81–84], providing important insights into rhythm-based formulations of Bayesian PC[22,25,28] and their use in affective disorders.

Encoding of unsigned pwPEs about stimulus outcomes was associated with dampened alpha/beta oscillations (10–16 Hz) in the cACC in HTA relative to LTA. This effect emerged at 500 and between 1200 and 1400 ms, converging with the latency of beta modulations during pwPE encoding in our previous studies of decision making and motor learning in state anxiety[33,34]. Temporary anxiety states, however, enhance the amplitude of beta oscillations[33,34]. The different direction of the alpha/beta modulation by pwPEs in trait and state[34] anxiety can be explained by the opposing patterns of computational results in both conditions: in ref. [34]. state anxiety was associated with a slower updating of beliefs about stimulus-outcome contingencies, which converges with the observed greater alpha/beta activity during encoding $|\varepsilon_2|$. Here, trait anxiety speeded belief updating on level 2 with corresponding suppression of 10–16 Hz activity. Importantly, in the present study, the alpha/beta attenuation effect was accompanied by a pronounced phasic increase in the amplitude of gamma responses in HTA at ~1 and 1.6 s. The relative gamma increase in HTA was identified across the cACC, lOFC, and dmPFC (label SFG in the DKT atlas[85]). The results are consistent with the notion that bottom-up PEs are encoded in gamma frequency oscillations and paralleled by downregulation of alpha/beta activity[19,22,25,29]. In the context of trait anxiety, the results align with the computational findings on uncertainty estimates, suggesting that trait anxiety promotes outcome-driven processing, enhancing the role of PEs in updating predictions[20,24].

Our study is the first to demonstrate that alterations in Bayesian belief updating during reward-based learning in trait anxiety are associated with changes in gamma activity across brain regions of the anxiety and decision-making networks. The ACC and medial PFC have been consistently shown to be involved in pathological and induced anxiety but also decision making and the processing of rewards[1,43,45,46,73]. The gamma effects we observed had very similar latencies in the cACC and the dmPFC, whereas no effects were found in the mOFC, which is considered to include the vmPFC in the anatomical parcellations in MEG studies[86,87]. In addition, we found that larger gamma activity across the cACC and dmPFC regions was associated with greater unsigned pwPEs, and trials with larger $|\varepsilon_2|$ were more likely to be followed by a response shift, more prominently in HTA. These findings are consistent with accounts of ACC and dmPFC function, suggesting that signals in these brain regions guide response choices[76,77]. The gamma effects in the dmPFC are also aligned with recent work linking gamma oscillations in the human dmPFC to encoding unsigned reward PEs during exploration-exploitation[32]. Our results provide preliminary evidence that aberrant encoding of pwPE via gamma oscillatory changes in dmPFC and cACC can account for behavioural alterations in affective disorders.

The antithetic modulation of alpha/beta and gamma activity by the unsigned pwPE regressor in the cACC converges with the vast evidence that increased gamma power in cortex during bottom-up processing is accompanied by a dampening of alpha/beta oscillations[25,35,88,89]. The lOFC, however, elicited a relative increase both in the gamma band and subsequently at alpha/beta frequencies (10–22 Hz). The lOFC plays a role in encoding

punishment value, nonreward and unpleasantness[48,90,91]. The relative HTA-LTA increase in alpha/beta activity in the lOFC to the pwPE regressor was paralleled by a negative amplitude change in the same time-frequency range to the lose regressor. It is unclear whether the increase in alpha/beta activity in the lOFC could be excitatory, contributing to further encoding the unsigned pwPE regressor. This interpretation remains speculative, although there is some evidence that beta activity may be excitatory in some brain regions during encoding unpredicted inputs, as shown recently in the primate parietal cortex[25]. The pwPE results across our ROIs highlight that the anticorrelated nature of gamma and alpha/beta oscillations during encoding pwPEs is expressed in specific regions of the decision-making networks (here the cACC).

The main convolution model additionally demonstrated a consistent HTA-related attenuation of alpha/beta activity during encoding precision weights on level 2 in the cACC. In the HGF for binary outcomes, the precision-weight term scaling the influence of PEs on the update of beliefs about the stimulus-reward contingency is simply $\sigma_2$, the expectation on informational uncertainty. Given that HTA increased $\sigma_2$, the GLM results thus associate greater precision weights driving belief updating in HTA with a reduction of alpha/beta activity in the cACC. This outcome could be mediated by increases in synaptic gain, as proposed for alpha oscillations in attentional tasks[24], which would promote the transmission of PEs, in line with our gamma results. This interpretation is supported by the latency of the effects, emerging at 1 and 1.6–1.7 s post-outcome, which closely matches the latency of the gamma-band effects in the cACC. In the lateral OFC, $\sigma_2$ was associated with greater 12–16 Hz activity around 0.3 and 1.5 s in high relative to low trait anxiety, thus converging with the relative increases in alpha/beta activity in this region for the pwPE regressor.

Last, a strong modulation of beta-band TF responses was found for uncertainty about volatility, $\sigma_3$, which weights the updates of beliefs on level 3. This effect emerged in the dmPFC and was associated with a pronounced relative increase in beta activity around 0.4 and 1.7 s. Our computational analyses did not find significant between-group differences concerning the expected uncertainty about volatility, which contributes to belief updates on level 3. Rather, the posterior mean on volatility had an initially higher estimate in HTA and remained high throughout the task. Accordingly, the GLM results suggest that trait anxiety enhances beta activity in the dmPFC during encoding uncertainty about volatility, $\sigma_3$, potentially inhibiting the regulation of the overestimation of volatility over time.

Combined, the neural and computational results on uncertainty estimates $\sigma_2$ and $\sigma_3$, and their modulation of precision weights provide a coherent picture of the relevance of assessing precision signalling to identify routes through which subclinical trait anxiety can hinder learning, particularly when learning is embedded in an environment rich in volatility. Our results build on the mounting evidence on the role of precision in explaining altered learning in a whole suite of clinical conditions and symptoms, such as hallucinations in Parkinson's disease[41,92], schizophrenia, autism[38] and psychosis[42]. An exciting avenue of future research in anxiety would be the combination of MEG recordings with pharmacological interventions, to assess the modulatory effects of neurotransmitters (dopamine[42,55]; acetylcholine[93]; noradrenaline[94]) on the neural oscillatory correlates of precision.

## Methods

**Participants**. We recruited 39 participants (24 female, 15 male) aged between 18 and 36 years (mean 22.8, SEM 0.9) who completed the MEG and behavioural study. We additionally acquired individual T1-weighted anatomical magnetic resonance images (MRI, details below). All participants reported having normal or corrected-to-normal vision. Individuals were excluded if they had a history of psychiatric or neurological disease or head injury, and/or were on medication for anxiety or depression. Written informed consent was obtained from all participants before the experiment, and the experimental protocol was approved by the ethics committee of the Institutional Review Board of the National Research University Higher School of Economics in Moscow, Russia.

Our sample size was estimated using the behavioural and EEG data from our recent work on decision making in state anxiety[14,34]. In ref. [14], we observed a large effect size of state anxiety on the HGF model parameter $\omega_2$ (the low-level tonic log-volatility estimate; $\Delta = 0.75$, CI = [0.55, 0.90], non-parametric effect size estimator, range 0–1: "Statistics and Reproducibility"). The size of the effect of temporary anxiety states on the beta activity modulation to unsigned pwPEs in ref. [34]. was $\Delta = 0.73$, CI = [0.65, 0.81]. Here, MATLAB function sampsizepwr (two-tailed t-test) was evaluated in those $\omega_2$ and beta activity data to estimate the minimum sample size for a statistical power of 0.80, with an $\alpha$ of 0.05. This analysis resulted in a minimum of 16 participants in each group (high, low *state* anxiety). In the current MEG study, to account for trait anxiety potentially associated with a smaller effect size than state anxiety, we recruited 20% more participants than the estimated minimum sample size: 20 and 19 participants in the LTA and HTA groups, respectively.

**Assessment of anxiety**. Participants' trait anxiety level was measured twice using Spielberger's State-Trait Anxiety Inventory[56] (STAI, trait subscale X2, 20 itemts, score 0–80): one assessment prior to attending the experiment as a selection procedure, and one at the beginning of the experimental session (to validate the pre-screened level). Trait anxiety refers to a relatively stable metric of an individual's anxiety level derived from the self-reported frequency of anxiety from past experiences[1]. Trait anxiety in subclinical populations is commonly measured using the STAI trait subscale, a measure thought to reflect the general risk factor for an anxiety or affective disorder[1]. This scale taps into the overall exaggerated perspective of the world as threatening, providing a good measure of how frequently a person has experienced anxiety across their life[95].

We used the trait anxiety scores as a selection process to form the two experimental groups: low trait anxiety (LTA, defined as a STAI score below or equal to 36) and high trait anxiety (HTA, defined as a STAI score above 45). These values were selected to include the normative mean value in the working adult population as upper threshold in the LTA group[56] (36, SD 9). In addition, the HTA threshold value was informed by the cut-off point (>45) used to denote clinically significant anxiety in treatment studies in anxiety disorder patients[96,97]. Trait anxiety scores ranged between 24 and 65. The average anxiety scores for each group were 30.5 (LTA, SEM 0.8) and 51.7 (HTA, SEM 1.5), comparable to LTA/HTA group values in recent investigations of reversal learning in trait anxiety[59,60]. Importantly, the experimental groups were balanced in terms of age and sex. The HTA group (mean age 22.6, SEM = 1.1) consisted of 12 females, while the LTA group (mean age 23.7, SEM = 1.0) consisted of 12 females. In addition to the trait inventory, measures of self-reported state anxiety using the STAI state subscale (X1, 20 items, score 0–80) were taken prior to the experiment and after completing the experiment.

During performance, our participants were monitored for physiological changes in heart-rate variability (HRV) and high-frequency HRV, to control for potential confounding factors that could modulate task completion[14] (Supplementary Results). Physiological responses did not vary as a function of the group or task block, despite a group effect on state anxiety (Supplementary Fig. 2; Supplementary Results).

**Experimental design and task**. We used a between-subject experimental design with two anxiety groups: HTA and LTA. Participants performed a probabilistic binary reward-based learning task in a volatile learning setting[53–55] (Fig. 1). The session was split between an initial resting state block (R1: baseline) of five minutes and two experimental reward-learning task blocks consisting of a total 320 trials (block 1, 160 trials – block 2, 160 trials). During the baseline block we recorded continuous MEG and electrocardiography (ECG). In this phase, participants were told to try to relax and fixate on a central point of the screen with their eyes open.

Similarly to ref. [14], participants were informed that the total sum of all their rewarded points would translate into a monetary reward at the end of the experiment. The calculation for this remuneration was the total sum of winning points divided by six plus 400, given in Russian rubles ₽ (for example, 960 points pays 960/6 + 400 = 560₽).

For every trial, a blue and an orange stimulus were shown on the monitor. Their location was either to the right or left of the centre, randomly generated in each trial. The maximum time allowed for a response before the trial timed out was 1300 ms ± 125 ms. Responses here were given by pressing a button in a response box with either the left or right thumb (corresponding to selecting either the left or right image). After the participant made their choice, the selected image was outlined in bright green for 1000 ms (± 200 ms) to indicate their response. After, feedback of the trial outcome was provided (win, green; lose or no response, red) in the centre of the screen for 1900 ms (± 100 ms). To conclude a trial, a fixation cross was shown in the centre of the screen (1750 ms [± 250 ms]). Participants were told to select the image they believed would reward them to maximise reward across the

320 trials, and also to modify their selections in response to any inferred changes to their underlying probability. Prior to starting the experimental task blocks (blocks 1, 2), each participant performed 16 practice trials and filled out the first state anxiety report. Between the two experimental task blocks, participants rested for a short self-timed interval. After completing the second task block, participants filled out the second state anxiety report before finishing the experiment.

**Modelling behaviour: The Hierarchical Gaussian Filter**. To model behaviour, we used the Hierarchical Gaussian Filter[49,50] (HGF; version 6.0.0, open-source software available in TAPAS, http://www.translationalneuromodeling.org/tapas, see ref. [98]). This model describes hierarchically structured learning across various levels $(1,2,…,n)$ and trials $k$, corresponding to hidden states of the environment $x_1^{(k)}$, $x_2^{(k)},…, x_n^{(k)}$ and defined as coupled Gaussian random walks (Fig. 2a). On level 2, $x_2^{(k)}$ denotes the current true probabilistic mapping between stimulus and outcome. In our modelling approach an agent would also infer the rate of change of the tendency towards a contingency mapping, that is, the level of environmental volatility on trial $k$. This is represented by the hidden state $x_3^{(k)}$. In the following we drop the trial index $k$ for simplicity. The HGF model has been used widely to describe task responses in multiple learning contexts[54,55,65,71,82,84,99]. We used TAPAS in Matlab R2020b.

Variational Bayesian inversion of the model provides the trial-wise trajectories of the beliefs, which correspond to the posterior distribution of beliefs about $x_i$ $(i = 2, 3)$ and represented by their sufficient statistics: $\mu_i$ (mean) and $\sigma_i$ (variance or uncertainty; inverse of precision, $\pi_i$ see Fig. 2b). Formally, the update equations of the posterior estimates for level $i$ $(i = 2$ and 3) take the form given by Eq. (1). Equation (1) illustrates that updates in the posterior mean on level $i$, $\mu_i$, are proportional to the precision-weighted PE, denoted by $\varepsilon_i$.

As in our previous work[14], we utilised a generative perceptual model for binary outcomes termed the 3-level HGF[49]. The input to the model was the series of 320 outcomes and the participant's responses. Observed outcomes in trial $k$ were either $u^{(k)} = 1$ if the blue image was rewarded (orange stimulus unrewarded) or $u^{(k)} = 0$ if the blue stimulus was unrewarded (orange stimulus rewarded). Trial responses were defined as $y^{(k)} = 1$ if participants chose the blue image, while $y^{(k)} = 0$ corresponded to the choice of the orange image. In the 3-level HGF, the first level $x_1^{(k)}$ represents the true binary outcome in a trial $k$ (either blue or orange wins) and beliefs on this level feature expected (irreducible) uncertainty due to the probabilistic nature of the rewarded outcome. In the absence of observation noise, $u^{(k)} = x_1^{(k)}$. The second level $x_2^{(k)}$ represents the true tendency for either image (blue, orange) to be rewarding. And the third level $x_3^{(k)}$ represents the log-volatility or rate of change of reward tendencies. In the HGF update equations, the second and third level states, $x_2^{(k)}$ and $x_3^{(k)}$, are modelled as continuous variables evolving as Gaussian random walks coupled through their variance (inverse precision).

We paired the 3-level HGF perceptual model with two alternative response models that map participants beliefs to their decisions. Response model (i) from refs. [50,55]. is governed by a unit-square sigmoid function that maps the predictive probability $m^{(k)}$ for an outcome on trial $k$ onto the probabilities that the individual will choose response 1 or 0, $p(y^{(k)} = 1)$ and $p(y^{(k)} = 0)$, respectively:

$$p(y|m,\zeta) = \left(\frac{m^\zeta}{m^\zeta + (1-m)^\zeta}\right)^y \cdot \left(\frac{(1-m)^\zeta}{m^\zeta + (1-m)^\zeta}\right)^{(1-y)} \quad (3)$$

The trial index $k$ has been dropped from Eq. (3) for clarity. The predictive probability $m^{(k)}$ depends on the variables that the HGF is inferring. As observed in (3), choice probability is shaped by a free fixed (time-invariant) parameter $\zeta$ that can be interpreted as inverse decision noise: the sigmoid approaches a step function as $\zeta$ tends to infinity (for further detail see Eq. 18 in ref. [49])

Response model (ii) from ref. [65] also used a sigmoid function to map an agent's beliefs to decisions, yet in this case the inverse decision noise in Eq. (3) is a function of a time-varying quantity: the prediction of log-volatility:[65] $e^{-\mu_3^{(k-1)}}$, thus depending on the participant's trial-wise beliefs on volatility—termed HGF$_{\mu3}$.

As in our prior work[14], parameters $\omega_2$ and $\omega_3$ were estimated in each individual (3-level HGF and HGF$_{\mu3}$; for the 2-level HGF, $\omega_3$ was fixed; Table S1). The response model parameter $\zeta$ was also estimated in the 3-level and 2-level HGF models, while parameters $\mu_3^{(0)}$ and $\sigma_3^{(0)}$ were estimated in model HGF$_{\mu3}$ (Table S1). Simulations conducted to assess the accuracy of parameter estimation in the HGF models demonstrated that the most accurate estimation was for parameters $\omega_2$ and $\mu_3^{(0)}$, while $\omega_3$ was poorly recovered (Supplementary Results), as shown previously[14,99].

We direct the reader to the original methods papers for more details on the derivation of the perceptual model and equations of the HGF quantities used in this paper[49,65]. Using the prior parameter values (Table S1) and series of inputs, maximum-a-posteriori (MAP) estimates of model parameters were then quantified and optimised using the quasi-Newton optimisation algorithm[65,98].

Model comparison at the population level was performed using random-effects Bayesian model selection[68] (BMS), as in previous work[14,54,55,65], using code from the MACS toolbox[69]. The BMS approach proposed by ref. [68] treats models as random effects that could vary across participants, but also have a fixed distribution in the population. Here, BMS was conducted using the individual log-model evidence (LME) values in each participant and model. The LME of a model is

negative surprise about the data, given the model, and measures the trade-off between a model's accuracy (fit) and complexity[55]. See Fig. 2c, d.

**Acquisition and preprocessing of MEG and ECG data**. Neuromagnetic brain activity was recorded using a 306-channel MEG system (102 magnetometers and 204 gradiometers, Elekta Neuromag VectorView, Helsinki, Finland) in sitting position. We used a head-position indicator to control for head movements, with four coils affixed to the head, two placed on the top of each side of the forehead, and two on the mastoid process of each side. Eye movements were controlled using an electrooculogram (EOG): Two horizontal EOG electrodes were placed each side of the temple, while the two vertical EOG electrodes were placed above and below one eye. In addition, two electrodes were used for electrocardiography (ECG) recording using in a two-lead configuration montage[100]. MEG, EOG, and ECG signals were recorded with a sampling rate of 1000 Hz and a band-pass filter of 0.1–330 Hz. Following the MEG acquisition phase, we de-noised the signals and corrected head movements using the Temporally extended Signal-Space Separation (tSSS) method[101], built-in in the Elekta software (Maxfilter™; Elektra Neuroscience 2010; settings: sliding window = 10 s, subspace correlation threshold = 0.9).

Further preprocessing of the MEG data (magnetometers and planar gradiometers) was conducted with the MNE-python toolbox[102] (Python version 3.9.4), as well as additional custom Python scripts (uploaded to the Open Science Framework, https://osf.io/wsjgk/). For the analysis of heart rate variability (Supplementary Fig. 2), the ECG signal was pre-processed using the FieldTrip toolbox[103] for MATLAB® (v. 2020b, The MathWorks, Natick, MA; Supplementary Results).

The MEG signals were downsampled to 250 Hz. Next, we removed power-line noise by applying a zero-phase notch filter at 50 and 100 Hz and removed biological artefacts (eye movements, blinks, heartbeats) using independent components analysis (ICA, fastICA algorithm). MEG signals that exceeded a certain amplitude threshold ($5^{-12}$ T for magnetometers, $4^{-10}$ T/cm for gradiometers) were excluded from further analysis. We also used the standard MNE-python algorithm for automatic detection of ICs relating to EOG and ECG artifacts, which were, however, validated visually in each subject. On average, we removed 4.5 components (SEM 0.1).

**Structural magneto resonance imaging**. Structural brain MRIs (1 mm3 T1-weighted) were obtained for all participants and used for source reconstruction. The MRI image was derived from a 1.5 T Optima MR 360 system (Spin Echo sequence, slice thickness 1 mm, field of view 288 × 288, TR = 600, TE = 13.5).

**Source analysis**. Source localisation of the MEG signals (combined planar gradiometers and magnetometers) was performed using Linearly Constrained Minimum Variance beamformers[52] in MNE-Python[102]. First, we used the individual T1-weighted MRI images to construct automatic surface-based cortical parcellations in each hemisphere with Freesurfer 6.0 software[104,105] (http://surfer.nmr.mgh.harvard.edu/). We chose the label map of the Desikan–Killiany–Tourville atlas[75] (DKT), which parcellates the cerebral cortex into 68 regions of interest (ROIs). Subcortical parcellations were also generated as default in Freesurfer but were not used in this study. Coregistration of the MR and MEG coordinate systems was performed with an automated algorithm in MNE-python available in the MNE software (mne_analyze: http://www.martinos.org/mne/stable/index.html). The coregistration step used the HPIs and the digitised points on the head surface (Fastrak Polhemus). We additionally verified that the coregistration of three anatomical (fiducial) locations (the left and right preauricular points and the nasion) were correct in both coordinate systems.

For forward model calculations, we used the command-line tool "mne watershed" to compute boundary element conductivity models (BEM) for each participant and selected the inner skull surface as volume conductor geometry. Then, we created a surface-based source space with "oct6" resolution, leading to 4098 locations (vertices) per hemisphere with an average nearest-neighbour distance of 4.9 mm.

For inverse calculations, LCMV beamformers were used. The adaptive spatial filters were computed with a data-covariance matrix in the target interval (0–1.8 s in outcome-locked and stimulus-locked analyses) and a noise-covariance matrix in a time interval preceding the stimulus (−1 to 0 s pre-stimulus) and outcome events (−3 to −2 s pre-outcome, thus corresponding to a waiting period before the stimulus). The regularisation parameter λ was set to 5%. To assess modulations in alpha (8–12 Hz) and beta (13–30 Hz) activity, the MEG data was band-pass filtered between 1–40 Hz prior to beamforming; source-level modulation of gamma activity (32–100 Hz) was evaluated using LCMV after applying a band-pass filter between 30 and 124 Hz (below the Nyquist rate at 125 Hz).

Last, source estimate time courses for individual vertices were obtained for a set of cortical labels corresponding to our ROIs: (1) rostral and caudal ACC (rACC, cACC; Fig. 3a); (2) lateral and medial OFC, which include the vmPFC according to some MEG studies[86,87] (Fig. 3b; but see ref. [106] for a debate on the vmPFC delineation); (3) superior frontal gyrus (SFG), representing the dmPFC[85] (Fig. 3b). In additional exploratory analyses, however, we conducted the analysis in the other labels of the DKT atlas to identify effects outside of our ROIs. The representative time course per label was obtained using the "PCA flip" method in MNE-Python.

This method consists of applying singular value decomposition to each vertex-related time course in the label, followed by extraction of the first right singular vector. Next, each vertex's time course is scaled and sign flipped. Following this procedure, we obtained five bilateral (10 in total) time courses corresponding with our three ROIs. An additional exploratory analysis was carried out in the other labels of the DKT atlas to identify effects outside of our ROIs.

**Spectral analysis and convolution modelling**. We estimated standard time-frequency representations of the source-level time series using Morlet wavelets. TF spectral power was extracted between 8 and 100 Hz. For alpha (8–12 Hz) and beta (13–30 Hz) frequency ranges we used 5–cycle wavelets shifted every sampled point in bins of 2 Hz. For gamma-band activity (32–100 Hz), 7-cycle wavelets sampled in steps of 2 Hz were used.

After transforming the source-level time series to TF representations, we used linear convolution modelling for oscillatory responses[51]. This approach is a frequency-domain version of similar approaches used in time-domain EEG analysis, such as the massive univariate deconvolution analysis[107]. Convolution modelling was implemented in SPM 12 (http://www.fil.ion.ucl.ac.uk/spm/) by adapting code developed by ref. [70] freely available at https://github.com/bernspitz/convolution-models-MEEG. This method allowed us to model the pseudo-continuous TF data resulting from concatenated epochs as a linear combination of explanatory variables (parametric HGF regressors or discrete stimulus, response and outcome regressors) and residual noise. The general linear model explains this linear combination as follows[51]:

$$Y = X\beta + \varepsilon \tag{4}$$

here $Y \in \mathbb{R}^{t \times f}$ denotes the measured signal, the TF transformation of the pseudo-continuous time series, and is defined over $t$ time bins (trials x peri-event bins in our study) and $f$ frequencies. The linear combination of $n$ explanatory variables or regressors is defined in matrix $X \in \mathbb{R}^{t \times n}$, and modulated by the regression coefficients $\beta \in \mathbb{R}^{n \times f}$. The noise matrix is denoted by $\varepsilon \in \mathbb{R}^{t \times f}$. Matrix $X$ is specified as the convolution of an input function, encoding the presence and value of discrete or parametric events for each regressor and time bin, and a Fourier basis function. This problem is solved by finding TF images $R_i$ for a specific type of event $i$ (e.g., outcome or response event type):

$$R_i = B\beta_i \tag{5}$$

In the expression above, $B$ denotes a family of $m$ basis functions (sines, cosines) over $p$ peri-event intervals, $B \in \mathbb{R}^{p \times m}$. This family is convolved with $k$ input functions $U$, representing the events of interest at their onset latencies ($U \in \mathbb{R}^{t \times k}$), to create the regressor variables $X$. Thus, $X = UB$. Using ordinary or weighted least squares, the predictors $\beta_i$ are estimated over frequencies and basis functions for each regressor $i$. The TF response images $R_i \in \mathbb{R}^{p \times f}$ have dimensions $p$ and $f$, and represent an impulse response function for a specific event. This TF image has arbitrary units and can be interpreted as deconvolved TF responses to the event types and associated parametric regressors. The TF images $R_i$ can be used for standard statistical analysis (see further details in ref. [51]). A schematic of the convolution modelling approach is presented in Fig. 6.

To adhere to the GLM error assumptions[51] we first converted the spectral power to amplitude by applying a square-root transformation. Our trial-wise explanatory variables included discrete regressors coding for stimuli (blue image left, blue image right), responses (right, left, no response), outcome (win, lose, no response) and relevant parametric HGF regressors. For computational efficiency, we conducted separate GLMs for outcome-locked and (exploratory) stimulus-locked analyses, inserting the relevant discrete and parametric regressors at the corresponding latencies in each case (Supplementary Fig. 4).

Our primary convolution model aimed to assess the parametric effect of pwPEs and precision weight terms ($\sigma_2$ and $\sigma_3$, see "Results") on TF responses in 8–100 Hz in a relevant time interval following the outcome event. In this GLM, similarly to ref. [14], we found high linear correlations between the absolute value of the second-level pwPEs, $|\varepsilon_2|$, and the third-level pwPEs about environmental uncertainty ($\varepsilon_3$; the Pearson correlation coefficients ranged from 0.67 to 0.95 among all 39 participants). Due to multicollinearity of regressors, pwPEs on level 3 have been excluded from subsequent analysis[14,72]. We chose the absolute value of $\varepsilon_2$ because its sign is arbitrary: the quantity $x_2$ is related to the tendency of one choice (e.g., blue stimulus) to be rewarding ($x_1 = 1$); yet this choice and therefore the sign of $\varepsilon_2$ on this level is arbitrary[14]. This GLM was estimated using a window from −0.5 to 1.8 s relative to the outcome event (outcome-locked analysis; Supplementary Fig. 4). The subsequent statistical analysis focused on the interval 0.2–1.8 s, informed by our previous work in state anxiety[34].

Last, in an exploratory analysis of the neural correlates of predictions within 8–30 Hz, we choose the absolute values of predictions on level 2 $|\hat{\mu}_2|$ and excluded the third level log-volatility predictions $\hat{\mu}_3$. This decision was also grounded on multicollinearity of regressors: There were high linear correlations between $|\hat{\mu}_2|$ and log-volatility $\hat{\mu}_3$ (Pearson r between −0.95 and 0.37, N = 39). As for pwPEs updating level 2, the sign of $\hat{\mu}_2$ is arbitrary as it represents the tendency of the stimulus-reward mapping for an arbitrary stimulus (e.g., mapping for the blue image). The absolute values $|\hat{\mu}_2|$ represent the strength of a prediction about the tendency towards a particular stimulus-reward contingency. Accordingly, if a

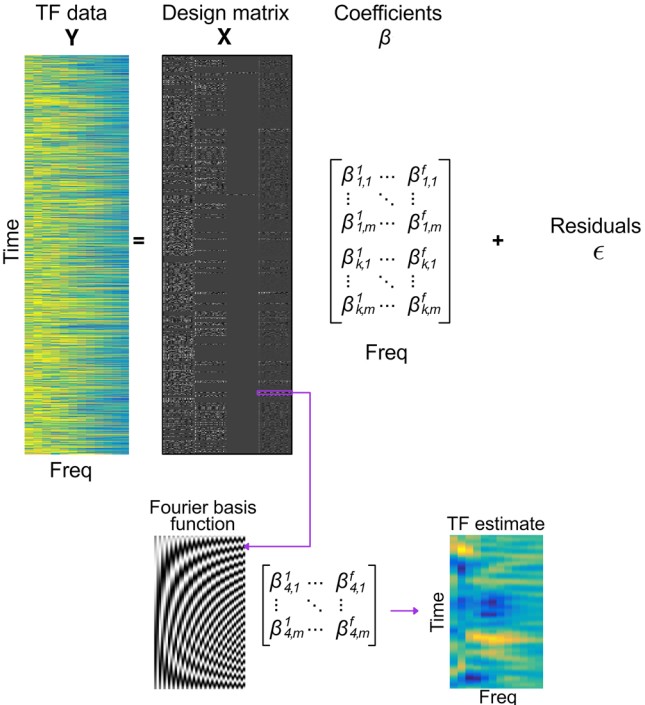

**Fig. 6 Convolution general linear model.** Standard pseudo-continuous time-frequency (TF) representations of the source-level MEG signal (Y) were estimated using Morlet wavelets. In GLM, signals Y are explained by a linear combination of explanatory variables or regressors in matrix X, modulated by the regression coefficients β, and with an added noise term (ε). The design matrix X shown in this figure was constructed by convolving Fourier functions (m = 20; 20 sine, 20 cosine functions; left inset at the bottom) with a set of input functions representing the onsets and value of relevant discrete or parametric events. In this example, we used k = 6 regressors (columns left to right): Outcome Win, Outcome Lose, Outcome No Response, absolute pwPE on level 2, informational uncertainty about the stimulus outcomes (σ₂), and uncertainty on level 3 (σ₃); all these regressors were defined over time. Solving a convolution GLM provides *response images* (TF estimate in the figure, arbitrary unit) that are the combination of the basis functions *m* and the regression coefficients βᵢ for a particular regressor type *i* defined over frequencies *f* and basis functions *m*. Thus, convolution GLM effectively estimates deconvolved TF responses (TF estimate, rightmost image at the bottom) to the event types and associated parametric regressors.

participant has a greater value of $|\hat{\mu}_2|$ in one trial, she will have a stronger expectation that given the correct stimulus choice a reward will be received. In this GLM we included, as additional discrete regressors, the stimuli (blue right, blue left), response (press left, press right, no response), and outcome (lose, win, no response) events. This model was estimated from −0.5 to 1.8 s around the stimulus event (stimulus-locked analysis Supplementary Fig. 4), yet this interval was refined in the subsequent statistical analysis (100–700 ms; see below).

In all convolution analyses, each discrete and parametric regressor was convolved with a 20th-order Fourier basis set (40 basis functions, 20 sines and 20 cosines). This setting allowed the GLM to resolve modulations of TF responses up to ~8.7 Hz (20 cycles/2.3 s; or ~115 ms). In an additional control analysis, we used a 40th-order Fourier basis set to assess gamma activity modulations by the unsigned pwPE regressor (Supplementary Fig. 3). This set provided a temporal resolution of 57.5 ms.

**Statistics and reproducibility**. Details on sample size estimation are provided in subsection "Participants". Statistical analysis of standard behavioural and computational model variables focused on between-group contrasts (LTA, HTA) after collapsing the block information. However, because ref. [14] demonstrated a large effect of the task block on behavioural win rates in state anxiety, we assessed this variable as a function of the Group (LTA, HTA) and Block (1, 2) factors.

Our dependent variables (DVs) were (i) win rates; (ii) win-stay/lose-shift rates, total switch rates; (iii) HGF trajectories averaged across trials in each task block

separately: (a) informational uncertainty about the stimulus outcomes ($\sigma_2$); (b) initial estimate on volatility ($\mu_3^{(0)}$), mean of the posterior distribution of beliefs about volatility ($\mu_3$), and the associated posterior uncertainty (variance, $\sigma_3$); (c) environmental uncertainty: $\exp(\kappa\mu_3^{(k-1)} + \omega_2)$, which is greater if the environment is more volatile; (iv) HGF perceptual model parameter quantities $\omega_2$ and $\omega_3$. Between-group comparisons of DVs ii–iv were carried out using pair-wise permutation tests (5000 permutations). We conducted a $2 \times 2$ Group $\times$ Block factorial analysis of the win rate (i). This was implemented using non-parametric factorial synchronised rearrangements[58] with 5000 permutations.

To address the multiple comparisons problem, where it arises (e.g., several post-hoc analyses), we control the false discovery rate (FDR) using an adaptive linear step-up procedure[108] set to a level of $q = 0.05$ providing an adapted threshold p-value ($P_{FDR}$). In the case of pair-wise statistical analyses we provide estimates of the non-parametric effect sizes for pair-wise comparisons and associated bootstrapped confidence intervals[109,110]. The within-group effect sizes are estimated as the probability of superiority for dependent samples ($\Delta_{dep}$), while the between-group effect sizes are based on the probability of superiority[109] ($\Delta$). Our results can be reproduced using code and data available at deposited in the Open Science Framework Data Repository under the accession code wsjgk.

Statistical analysis of the source-level TF responses obtained in convolution modelling was performed with the FieldTrip Toolbox[103], after converting the SPM TF images (in arbitrary units, a.u.) to a Fieldtrip structure. Given the large inter-individual differences typically observed in the amplitudes of MEG neuromagnetic responses, the source-level TF images were baseline corrected by subtracting the average baseline level ($-300$ to $-50$ ms) and dividing by the baseline standard deviation (SD) of the interval. We used a cluster-based permutation approach[103,111] (two-sided t-test, 1000 iterations) to assess between-group differences in TF responses across 10 anatomical labels, time points, and frequency bins (8–100 Hz for the outcome-locked GLM model; 8–30 Hz for the exploratory stimulus-locked prediction GLM model). We did not consider spatial relations between anatomical labels but focused on spectrotemporal clusters. Based on the latency of the effects in our previous work[34], we chose as the temporal intervals of interest for the statistical analysis 200–1800 ms for the outcome-locked convolution models, and 100–700 ms for the stimulus-locked GLM. This analysis controlled the family-wise error rate (FWER) at level 0.025 (exploratory uncorrected results will be explicitly stated).

**Reporting summary**. Further information on research design is available in the Nature Portfolio Reporting Summary linked to this article.

## Data availability

The data that support findings of this study are available from the Open Science Framework Data Repository under the accession code wsjgk.

## Code availability

Code for the source reconstruction analysis (MNE Python) and convolution modelling (Matlab / SPM) has been deposited in the Open Science Framework Data Repository under the accession code wsjgk.

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

## Acknowledgements

The study was supported by Goldsmiths University of London, funded by the Economic and Social Research Council (ESRC) and the South East Network for Social Sciences (SeNSS) through grant ES/P00072X/1. MHR, VVN and TF were partially supported by the Basic Research Programme of the National Research University Higher School of Economics (Russian Federation). The research used the Elekta Neuromag 306-channel MEG system at Centre for the neurocognitive research (MEG-Centre) in Moscow (Russian Federation) during 2020–2021.

## Author contributions

T.H. contributed to methodology, software, formal analysis, visualisation, writing-original draft preparation, and funding acquisition. M.H.R. contributed to conceptualisation, methodology, software, formal analysis, visualisation, writing-original draft preparation, writing-reviewing and editing, supervision, funding acquisition. Z.G. and M.I. contributed to conceptualisation, investigation. T.F. contributed to conceptualisation, writing-reviewing and editing. V.V.N. contributed to conceptualisation, writing-reviewing and editing, funding acquisition.

## Competing interests

The authors declare no competing interests.
