## [Peer Review File · Communications Biology]

Reviewers' comments:

Reviewer #2 (Remarks to the Author):

The authors conducted a probabilistic reversal learning experiment where they collected behavioral as well as MEG and structural MRI data. Two groups with high and low trait anxiety were sampled and compared. The authors fitted a hierarchical Gaussian filter model to the behavioral data and found that high trait anxiety individuals switch more often, and over-estimate volatility as well as uncertainty compared to low trait anxiety individuals. Using the unsigned precision-weighted prediction errors from the model as a regressor for analyzing the MEG data, they found alterations in gamma as well as alpha/beta band activity in decision making and reward processing areas during encoding of prediction errors. The authors interpret the findings as high trait anxiety individuals updating their beliefs more strongly due to the higher volatility estimate, which leads to larger prediction errors, which are encoded throughout the brain in the gamma band in relevant decision making areas.

The work is methodologically sound and the results and the conclusions are convincing. However writing and structure can be improved as it was at times hard to understand what exactly the authors did without going back and forth in the manuscript and reading the authors previous work. This is in part due to an unclear writing style, the "methods after results" structure, and due to lacking context from other work. Sometimes I am unsure which is which but I will try to tease it apart as well as I can so that the authors can improve their manuscript, see points 1-3.

Firstly, could the authors please add page and line numbers for easier reference to specific parts of the manuscript? I will refer to page numbers as the number of the page my PDF viewer shows (and hope to not be off).

Major comments:

1. In general, the results part reads as if I should have read the methods part before. Hence it would be good if the authors would revise the order in which they give information in the "results before methods" structure and explain a bit more detail in the results part, and give the information needed to understand the results without needing to read the methods first. In particular, I did not understand the exact experimental procedure nor the behavioral modeling and model comparison fully. Here are a couple of examples, although it may be that I did not catch all of them:

- it is not clear how the reward probability trajectory $p(\text{reward}|\text{orange})$ was generated and if it was the same for all participants or not.
- Similarly, the authors use variable names and abbreviations in the text and figures, which are only explained in the methods, e.g. TBn. I thought they meant the first contingency block, but in fact I think it is explained later that the experiment was actually divided in two parts to which TB refers. It was not clear that this was a part of the experimental procedure.
- The results part has many variable names in the text and figure captions, without them having been comprehensively introduced, which makes the interpretation of the results difficult, e.g. epsilon on page 9 (introduction in the text comes on page 11, but it is then not explained in the methods where the model is explained).
- It is also unclear why certain analyses were made (see point 2), especially the model comparison, which is not referenced in the text in the results section, only later on page 25 (see point 4).
- It was not clear what the difference between the models is, and how perceptual and behavioral model are combined (see point 2). It would help if the model names are explicitly stated in the text when the models are first introduced (page 7, second to last paragraph).

I understand that one can not give all the details in this type of structure, but then the figures and results should be presented in a way that they are understandable and make sense even without knowing all procedures in detail. For example, without having a representation of the equations in mind, all the variable names are not very helpful, and could be left out maybe, or given other names?

I am a bit torn on how to resolve this. On one hand giving more information earlier would work, but on the other hand maybe having even more information in the results is not feasible in this structure, but the figures and text could instead be a bit more “zoomed out” to a birds view so that it is OK to not understand all the methods in detail at this point?

2. Results and methods are sometimes difficult to understand without the context of the authors previous work (Hein et al. 2021) and some references used.

- The behavioral model(s) are not explained fully mathematically, which makes understanding the analysis in depth difficult. It would be good to show at least some of the equations, if only in a supplementary file, or explicitly refer the reader to the authors previous work (Hein et. al. 2021) in the respective methods sections (pages 23&24). The gist of the perceptual model can be understood from the graph in Figure 2, so these equations don't necessarily have to be all typed out, but please at least show how the behavioral model concretely connects perception and choices, and importantly the equation for ϵ_2 in the methods part, which is needed to understand the MEG results. I actually had to refer to the authors' previous work to understand which three models are compared in the results, and I had initially interpreted the text in the results part very differently and had thought that all three models are perceptual models and each is connected with one of two choice models, which would then lead to comparing 6 models (see point 1 also).
- Could you please provide an explicit rationale for the choice of the choice function of HGF μ_3 in a sentence or two, as to why it is reasonable to assume an adaptive decision temperature that depends on the log volatility (on page 9 for example). Why not use other variables like the volatility etc.
- Similarly for the experiment. I am actually still unsure how exactly the experiment looked like, which as far as I understand was also used in the authors previous work (Hein et. al. 2021). Please be more explicit in the explanations (see also point 3).
- Similarly for the explanation of the GLM, for which I looked into the authors' previous work (see also points 3 & 5).
- Why the model comparison (page 9) between the two families and the 5 concrete models was chosen at all was also unclear from the text. In fact the model comparison is not referenced in the main text in the results part (only later in the methods on page 25). It is just in Figure 2 and raised more questions than it answered (see point 4). Specifically, the authors cite (de Berker et al. 2016) but do not provide any reasoning themselves. Please provide some reasoning so that the reader has context.

It is clear that one cannot always reiterate all previous work in detail in a new article, however each paper should aim to be a self contained.

3. The writing could be more clear and precise. In some parts where I referred to the authors previous work, I think the same information may have been given, but unfortunately not as clearly explained, e.g. in the GLM explanation in the methods, as well as in the behavioral modeling part of the methods.

4. I find the model comparison of the behavioral models odd, as the authors do not provide reasoning as to why it is needed here, and why the concrete two RL models were chosen. I checked the reference the authors give (de Berker et al. 2016), which interestingly also does not provide a reasoning on why exactly these models are compared. De Berker et al. (2016) only refer to other work (Iglesias et al. 2013), which finally states that they wanted to make sure that the participants do in fact employ the “more complex” hierarchical model and don't use a simpler tracking of action values. Maybe it is a standard procedure I am unaware of, and if so I apologize. However, nowadays it seems not very controversial to assume that humans employ a hierarchical Bayesian model to update learning rates based on environmental volatility, as shown in many works the authors cite, hence I am not sure if a model comparison is needed at all. If you do want to leave it in, please provide an explicit explanation of the reasoning for it and why the two competing models RL were chosen.

As a tangent, if one wanted to compare the model to reinforcement learning models, there are other

interesting models one could pitch the HGF against. In fact, seeing the model comparison the authors showed made me very curious how the model would compare. There is for example a mood RL model, that uses mood as a momentum (Eldar et al. 2016). There are also RL models (for example Bai et al. 2014) with adaptive learning rates based on volatility, which may yield similar findings in regards to group differences. There are surely many more new interesting models out there. This is not necessary for this work, but could be interesting for some follow up.

5. Could the authors explain the “model-based” MEG analysis in more detail. Page 28 only specifies “relevant parametric HGF regressors”. Which are those? All HGF variables, a subset, or only the unsigned pwPE (and later its components)? This is in my opinion important information to understand the results figures in depth. If I read the text and the authors previous work correctly, I think that only ϵ_2 was added to the GLM. Why were other quantities not added? I can see that it makes sense to find correlates of this more so than for other variables. However since it seems to be the only behavioral model variable that was added, it begs the question whether correlations with this in the MEG signal are in fact correlations with the variable specifically, or if they are general correlations to the behavior and model. The high collinearity of ϵ_2 and ϵ_3 could speak for that.

6. Why are alpha and beta analyzed together but independently from gamma (page 26)? Why not make all three independent or analyze all three together? I wonder if this introduces correlations or biases in the analysis. If there are references for this approach, please give them here. If it is due to the physiological dissociation the authors mention in the introduction (page 3), please state this at the point point of the methods where the band-pass filter is mentioned.

Minor comments:

Page 4 (bottom, beginning of results section):

- in the description of the participant samples in the results no specifics of the two groups are given, specifically no group sizes, these are only given for the whole group, and since the total numbers of the participants is odd the matching was probably not 1 to 1.
- could the authors specify in the text how exactly the reward probability trajectory $p(\text{reward}|\text{orange})$ was generated? In the caption to Figure 1B it seems to be that the trajectory was randomized and different for each participant. If this is the case, could the authors please also specify how they defined blocks, or which block lengths were chosen. And was this factored into the comparison between groups
- could the authors state before the second to last line (about the heart rate) which physiological quantities were assessed during the experiment?

Page 6 (figure caption):

- the abbreviation TB2 is mentioned in the caption but not explained. I assume it is for task block? Same in the x axis label in C and in the Supplementary figure 2.
- The figure caption speaks about inputs u but u is not shown in the graph of the model

Page 8:

- There is a bracket missing on the y axis label of Figure 2F.
- ϵ_i does not occur in the text and it is at this point unclear what it is.

Page 12:

- please add an x axis label to Figure 3 D,F,H

Page 18:

- in paragraph 2 it should read lose-shift not lose-shit rate :)

Page 25:

- can the authors please specify the content of the “custom python scripts” for preprocessing? Without knowing the content it is hard to judge if the preprocessing was sound or not, as anything could be done in a custom script.

1. de Berker AO, Rutledge RB, Mathys C, Marshall L, Cross GF, Dolan RJ, Bestmann S. 2016. Computations of uncertainty mediate acute stress responses in humans. *Nat Commun* 7:10996. doi:10.1038/ncomms10996
2. Iglesias, S., Mathys, C., Brodersen, K. H., Kasper, L., Piccirelli, M., den Ouden, H. E., & Stephan, K. E. (2013). Hierarchical prediction errors in midbrain and basal forebrain during sensory learning. *Neuron*, 80(2), 519-530.
3. Hein, T. P., de Fockert, J., & Ruiz, M. H. (2021). State anxiety biases estimates of uncertainty and impairs reward learning in volatile environments. *NeuroImage*, 224, 117424.
4. Eldar, E., Rutledge, R. B., Dolan, R. J., & Niv, Y. (2016). Mood as representation of momentum. *Trends in cognitive sciences*, 20(1), 15-24.
5. Bai, Y., Katahira, K., & Ohira, H. (2014). Dual learning processes underlying human decision-making in reversal learning tasks: functional significance and evidence from the model fit to human behavior. *Frontiers in psychology*, 5, 871.

Reviewer #3 (Remarks to the Author):

This study revealed that key brain regions of the anxiety and decision-making circuitry exhibit changes in oscillatory activity that can account for behavioural and computational effects of anxiety within a Bayesian predictive coding framework. In a magnetoencephalography experiment, they observed enhanced gamma activity in HTA during the encoding of unsigned pwPE, and these effects emerged in the ACC, dmPFC, and OFC. I found the paper to address a current topic and to be largely well-written. My main concerns with the paper relate to theoretical basis and analytical perspective, including the selection of model frameworks, and the literature review which seems incomplete with respect to more recent publications. My more detailed comments are given below. I hope the authors take these comments as constructive and helpful to improve this interesting paper.

1.The authors built on recent progress in rhythm-based formulations of Bayesian predictive coding to identify sources of oscillatory modulations associated with altered learning in a volatile environment in subclinical trait anxiety. It is not clear whether a hierarchical model is necessary. Also, the authors did not explore how other types of hierarchical models would perform. If you decide to keep only this hierarchical model, stronger justification is necessary.

2.In a Bayesian PC framework, this study revealed that HTA undermines learning through an overestimation of volatility leading to faster belief updating. The authors found the HTA individuals were significantly more uncertain about the environment. However, it's incomprehensible to me, no significant differences were found in uncertainty about volatility or the tonic learning rates at levels 3. In my opinion, the authors should explain these insignificant results in the discussion section.

3.Also, the authors employed a three-level hierarchical Bayesian model to test the rhythm-based formulations of Bayesian predictive coding, and revealed that the significant difference mainly comes from the third level. However, it's strange to only focus on the encoding of pwPEs at level 2 (e.g., unsigned precision-weighted prediction errors about stimulus outcomes, $|\epsilon_2|$). What are the results at level 3 (e.g., $\epsilon_3, \sigma_3, \mu_3$)?

4.The authors observed enhanced gamma activity and accompanied by suppression of alpha/beta activity during the encoding of pwPEs in HTA. These seems inconsistent with the author's previous study. In Hein et al. (2022), the authors found that state anxiety increased alpha/beta activity in frontal regions in processing of pwPEs. I wonder how to understand this potential inconsistency.

We would like to thank the reviewer for their feedback.

1. *In general, the results part reads as if I should have read the methods part before. Hence it would be good if the authors would revise the order in which they give information in the “results before methods” structure and explain a bit more detail in the results part, and give the information needed to understand the results without needing to read the methods first. In particular, I did not understand the exact experimental procedure nor the behavioral modeling and model comparison fully.*

We have provided additional details on methodological aspects in the Results section, to support the presentation of the results and further methodological details can be found in the Methods section at the end of the manuscript (in line with Communication Biology's instructions for authors) and in the Supplementary Methods. We asked the Senior Editor whether we could include the Methods section in full before the Results, but this was not possible as it would not align with the structure of the journal. We hope that the new methodological details embedded in Results are useful. We acknowledge that as a reviewer and methods-oriented reader, we also often feel the need to read the Methods sections before the Results sections in similar format journals.

2. *Firstly, could the authors please add page and line numbers for easier reference to specific parts of the manuscript?*

Thanks for pointing this out. We have introduced page numbers, which is in line with the Comm Biology format. The page numbers inferred by the reviewer were correct, but we hope the new explicit paging system is more efficient during the reviewing process.

3. *It is not clear how the reward probability trajectory $p(\text{reward}|\text{orange})$ was generated and if it was the same for all participants or not.*

We explain this aspect in more details now “The stimulus-outcome contingency mapping changed 10 times across the total 320 trials, and the possible contingencies were 0.9/0.1, 0.1/0.9, 0.7/0.3, 0.3/0.7, and 0.5/0.5”. We generated the order of these contingency phases pseudorandomly and separately for each participant, with the constraint that those five types of contingency mappings had to occur within each block. Each contingency mapping changed every 26-38 trials (pseudorandom selection). This information has now been included in pages 4-5 of Results and in Figure 1B. In addition, we provide now a Supplementary Figure displaying the contingency mappings for the group average and for each individual (**Supplementary Figure 1**). There were no between-group differences in the mapping across trials.

4. *Similarly, the authors use variable names and abbreviations in the text and figures, which are only explained in the methods, e.g. TBn. I thought they meant the first contingency block, but in fact I think it is explained later that the experiment was actually divided in two parts to which TB refers. It was not clear that this was a part of the experimental procedure.*

Thank you. We have removed TB and used Block 1 and Block 2 for the figures and text.

5. *Similarly for the experiment. I am actually still unsure how exactly the experiment looked like, which as far as I understand was also used in the authors previous work (Hein et. al. 2021). Please be more explicit in the explanations (see also point 3).*

We have now explicitly stated at the beginning of the Results section (page 4) and in Figure 1 that the task was a one-armed bandit task. We have also provided additional details on the task in Results (page 4; see also reviewer's point 3 above).

6. *The results part has many variable names in the text and figure captions, without them having been comprehensively introduced, which makes the interpretation of the results difficult, e.g. epsilon on page 9 (introduction in the text comes on page 11, but it is then not explained in the methods where the model is explained).*

Agreed. We introduce ϵ_i in the text in page 8, after we present the general form of the update equations for hidden states (eq. [1]). We have however removed the term ϵ from Figure 2 for simplicity.

7. *It is also unclear why certain analyses were made (see point 2), especially the model comparison, which is not referenced in the text in the results section, only later on page 25 (see point 4). It was not clear what the difference between the models is, and how perceptual and behavioral model are combined (see point 2). It would help if the model names are explicitly stated in the text when the models are first introduced (page 7, second to last paragraph).*

We now have explained more in detail the procedure and rationale for conducting model comparison, see page 9 in Results (and briefly on page 28 in Methods). We briefly introduce the approach and results in page 9 and the updated Figure 2DC (and figure caption) but included the new details in Methods. As in previous work with the HGF, we used random-effects Bayesian model selection (Stephan et al 2014; Soch et al., 2016). This approach allows us to use the log-model evidence obtained for each participant and model and to use these values to obtain two quantities:

- Exceedance probability: The probability that one model explains the data better than other models (probabilistic estimate).
- Expected frequency: Conditional estimates on how frequently one model wins against the other models (probabilistic estimate).

Following Soch et al (2016), when using model selection for methodological control, i.e. identifying the optimal model for data analysis, the optimal model is the one with the highest expected frequency, which reflects the estimated model frequency. On the other hand, when the purpose is discovery science, i.e. deciding between competing theories of brain function, the best model should outperform the others in terms of exceedance probabilities (Stephan et al., 2009, eq. 16). It is standard in HGF papers to obtain both quantities and choose the model that outperforms other models in both parameters.

Hierarchical Bayesian models like the ones of the HGF family have been shown to outperform standard RL models in all recent studies (Iglesias et al., 2013; Diaconescu et al., 2014; Aukstulewicz et al., 2017). Accordingly, arguably, one could skip them (as we have done now, see point 8 below). On the other hand, among the HGF perceptual and response models, it is not clear a priori which one will explain the data best. It was therefore essential to conduct Bayesian model selection.

We also now provide further details in the Results (pages 8-9) and Methods (pages 27-28) sections about our different response and perceptual models, why they were introduced, and how they were combined, and how they differ.

- Stephan KE, Penny WD, Daunizeau J, Moran RJ, Friston KJ. 2009. Bayesian model selection for group studies. *Neuroimage* 46:1004–1017. doi:10.1016/j.neuroimage.2009.03.025
- Soch J, Haynes JD, Allefeld C. How to avoid mismodelling in GLM-based fMRI data analysis: cross-validated Bayesian model selection. *NeuroImage*. 2016 Nov 1;141:469-89.

8. *Why the model comparison (page 9) between the two families and the 5 concrete models was chosen at all was also unclear from the text. In fact the model comparison is not referenced in the main text in the results part (only later in the methods on page 25). It is just in Figure 2 and raised more questions than it answered (see point 4). Specifically, the authors cite (de Berker et al. 2016) but do not provide any reasoning themselves. Please provide some reasoning so that the reader has context.*

Also: I find the model comparison of the behavioral models odd, as the authors do not provide reasoning as to why it is needed here, and why the concrete two RL models were chosen. I checked the reference the authors give (de Berker et al. 2016), which interestingly also does not provide a reasoning on why exactly these models are compared. De Berker et al. (2016) only refer to other work (Iglesias et al. 2013), which finally states that they wanted to make sure that the participants do in fact employ the “more complex” hierarchical model and don’t use a simpler tracking of action values. Maybe it is a standard procedure I am unaware of, and if so I apologize.

However, nowadays it seems not very controversial to assume that humans employ a hierarchical Bayesian model to update learning rates based on environmental volatility, as shown in many works the authors cite, hence I am not sure if a model comparison is needed at all. If you do want to leave it in, please provide an explicit explanation of the reasoning for it and why the two competing models RL were chosen.

Agreed. Studies using the HGF have typically chosen widely used RL models such as Rescorla-Wagner or Sutton Barto. As the reviewer suspects, this is indeed standard procedure in recent HGF papers. We agree it is indeed trivial to observe that HGF models will outperform the fairly simple RL models, as this is expected and indeed what every HGF study finds^{14,54,55,68}. We have clarified this aspect now in page 9 as an argument to focus on HGF models: “While previous HGF studies^{23,54,55}, including our own work¹⁴, also considered widely used and relatively simple reinforcement learning models, the model comparison

approaches consistently demonstrated that the HGF models described the data best ...

We were specifically interested in using the trajectories of beliefs obtained in the HGF (posterior mean and posterior variance) to assess the effect of trait anxiety on learning in this simple one-armed bandit task with volatility. It is interesting that in our previous work in state anxiety (Hein et al., 2021), the optimal model used the response model with fixed decision noise, while the current study in trait anxiety identified the best model as the one in which beliefs are mapped to responses as a function of dynamic volatility estimates. This very same model is also the best one in new study we are completing in trait anxiety using a motor learning task. In Results, we also comment on alternative Bayesian models we would like to assess in the future (page 9), with a caveat: “A *direct comparison between these different Bayesian models is not straightforward at this point as model inversion for the HGF uses variational Bayes, while the probabilistic model by Piray and Daw uses Monte Carlo sampling to estimate belief distributions*⁶⁷. *Reformulating models to the same Bayesian inference framework to allow for model comparison is challenging*⁶⁸ and is not feasible in this case.”

9. *As a tangent, if one wanted to compare the model to reinforcement learning models, there are other interesting models one could pitch the HGF against. In fact, seeing the model comparison the authors showed made me very curious how the model would compare. There is for example a mood RL model, that uses mood as a momentum (Eldar et al. 2016). There are also RL models (for example Bai et al. 2014) with adaptive learning rates based on volatility, which may yield similar findings in regards to group differences. There are surely many more new interesting models out there. This is not necessary for this work, but could be interesting for some follow up.*

Thank you for these suggestions. An additional important complementary line of work in mood disorders has primarily focused on using reinforcement learning models (e.g. Gagne et al., 2020; Aylward et al., 2019). These models are usually formulated in different frameworks to achieve model inversion, and integrating the HGF in these frameworks has rendered elusive so far (e.g. hBayesDM in R uses RL models but the package contributors did not implement the HGF so far).

- Aylward J, Valton V, Ahn W-Y, Bond RL, Dayan P, Roiser JP, Robinson OJ. 2019. Altered learning under uncertainty in unmedicated mood and anxiety disorders. *Nat Hum Behav* 3:1116–1123. doi:10.1038/s41562-019-0628-0
- Gagne C, Zika O, Dayan P, Bishop SJ. Impaired adaptation of learning to contingency volatility in internalizing psychopathology. *Elife*. 2020 Dec 22;9:e61387.

10. *I understand that one can not give all the details in this type of structure, but then the figures and results should be presented in a way that they are understandable and make sense even without knowing all procedures in detail. For example, without having a representation of the equations in mind, all the variable names are not very helpful, and could be left out maybe, or given other names?*

I am a bit torn on how to resolve this. On one hand giving more information earlier would work, but on the other hand maybe having even more information in the results is not feasible in this structure, but the figures and text could instead be a bit more “zoomed out” to a birds view so that it is OK to not understand all the methods in detail at this point?

Thank you. We were aware that a lot of methodological details would be essential to understand and critically review the results. We had tried to add some relevant details but we were also unsatisfied with it. We have tried to give a bit of more details now in the Results, while keeping additional details in Methods at the end of the manuscript in agreement with the Comm Biology formatting guidelines. We had looked at a few examples of HGF and neuroimaging work in journals with a similar Introduction/Results/Discussion/Methods structure and found that the authors included all details at the end in Methods or in Supp. Materials. However, we agree that our manuscript will benefit from a clearer – yet succinct-- presentation of key variables, formulas and methods during the Results section. See also our reply to points 1-6 above.

- 11. The behavioral model(s) are not explained fully mathematically, which makes understanding analysis in depth difficult. It would be good to show at least some of the equations, if only in a supplementary file, or explicitly refer the reader to the authors previous work (Hein et. al. 2021the respective methods sections (pages 23&24). The gist of the perceptual model can be understood from the graph in Figure 2, so these equations don't necessarily have to be all typeout, but please at least show how the behavioral model concretely connects perception and choices, and importantly the equation for epsilon_2 in the methods part, which is needed to understand the MEG results. I actually had to refer to the authors' previous work to understand which three models are compared in the results, and I had initially interpreted the text in the report very differently and had thought that all three models are perceptual models and each is connected with one of two choice models, which would then lead to comparing 6 models (see point 1 also).*

Agreed. We introduce the general form of the update equations for the hidden states in the perceptual HGF model in Results (eq. 1, page 8), while the general form of the response model is presented in Methods, in pages 27-28 (eq. 3). The two response models differ by using a time-invariant (zeta) or time-varying ($\exp(-\mu_3)$) quantity as the decision noise parameter modulating how decisions are mapped to beliefs. We also provide an intuition for the difference between these two response models in Results, pages 8-9.

- 12. Could you please provide an explicit rationale for the choice of the choice function of HGF μ_3 in a sentence or two, as to why it is reasonable to assume an adaptive decision temperature that depends on the log volatility (on page 9 for example). Why not use other variables like the volatility etc.*

We have added this to pages 8-9. The second response model depends on the expectation on log-volatility, $e^{-\mu_3^{k-1}}$, as variable μ_3 is estimated in the log space in the HGF.

GLM

13. *Similarly for the explanation of the GLM, for which I looked into the authors' previous work (see also points 3 & 5).*

We have rewritten the GLM/Results section, and introduced detailed information on the hypothesis-driven GLM analysis and our choice of regressors in Results, page 13. We have also created a scheme (**Supplementary Figure 4**) illustrating the hypothesised time windows encompassing the effect of HGF regressors on oscillatory neural activity. As recommended by the reviewer, we combine the relevant pwPE and precision weight terms as parametric regressors in a *single* outcome-locked GLM, and include the behavioural events as discrete regressors (as done previously). Exploratory analysis of predictions is conducted in a separate stimulus-locked GLM.

We display below two excerpts from page 13:

*“Based on prior work^{23,34}, we hypothesised that the neural responses correlated with pwPEs and precision terms would be observed in a time interval following the outcome presentation, whereas the effect of predictions before observing the outcome could be determined by analysing the post-stimulus (pre-outcome) interval^{23,34,42}. A scheme of the hypothesised timeline of effects is presented in **Supplementary Figure 4**. “*

...

*“The main outcome-locked GLM evaluated the effect of parametric regressors $|\varepsilon_2|$, σ_2 , and σ_3 , on the TF responses, while it controlled for the effect of discrete outcome events (win, lose, no response). Next, in an exploratory analysis, we implemented a stimulus-locked GLM to assess the neural oscillatory processes correlated with the parametric regressor $|\hat{\mu}_2|$. This model additionally included discrete regressors denoting the stimuli presentation (blue image on the left or right side), the participant's response (left, right, no response), and outcome cues (win, lose, no response) at their respective onset (**Methods**). “*

The specific details of the time-frequency analysis in convolution model approach remain in the Methods section as we considered these less crucial to understand the results, and were mindful of the formatting guidelines of Communication Biology.

14. *The writing could be more clear and precise. In some parts where I referred to the authors previous work, I think the same information may have been given, but unfortunately not as clearly explained, e.g. in the GLM explanation in the methods, as well as in the behavioral modelling part of the methods.*

Agreed. See our replies to similar points raised above. We hope to have tackled this aspect better in the revised manuscript.

15. *Could the authors explain the “model-based” MEG analysis in more detail. Page 28 only specifies “relevant parametric HGF regressors”. Which are those? All HGF variables, a subset, or only the unsigned pwPE (and later its components)? This is in my opinion*

important information to understand the results figures in depth. If I read the text and the authors previous work correctly, I think that only epsilon_2 was added to the GLM. Why were other quantities not added? I can see that it makes sense to find correlates of this more so than for other variables. However since it seems to be the only behavioral model variable that was added, it begs the question whether correlations with this in the MEG signal are in fact correlations with the variable specifically, or if they are general correlations to the behavior and model. The high collinearity of epsilon 2 and 3 could speak for that.

We have clarified our choice of HGF regressors for the MEG GLM analyses in the revised manuscript in page 13. We have also created a figure scheme (**Supplementary Figure 4**) indicating in which time windows our GLM analyses are conducted. In brief, different regressors can be included depending on the hypotheses (based on prior work) on the timeline of their effects. When assessing neural oscillatory effects locked to the outcome events (0-2 seconds after the outcome event), we hypothesise that pwPE values and precision weights will explain the neural changes. This is partly based on evidence from previous studies (Auksztulewicz et al., 2017, Haarsma et al., 2021). In Bayesian predictive coding, it is proposed that agents update their beliefs after receiving new information or observing outcomes through prediction errors weighted by precision (pwPE). In hierarchical models, such as the HGF, this would imply using the different types of pwPE trajectories, such as $|\epsilon_2|$ and ϵ_3 . Precision weights, which in the HGF are proportional to uncertainty (inverse precision) on each level, σ_2 and σ_3 , have also been associated with neural changes at the outcome onset (Haarsma et al., 2021) and are included in our main outcome-locked GLM. Thus, our main GLM model assessing the time-frequency responses after the outcome event include these regressors, as the most relevant predictors hypothesised to modulate brain activity in this window. See page 13 and our reply to query #13 above.

In the case of a one-armed bandit task, such as the one we used, ϵ_2 and ϵ_3 are not linearly correlated, despite their coupling function, due to the sign in ϵ_2 . However, we hypothesise that the neural representation of pwPE updating beliefs on the reward contingencies will reflect the absolute value of pwPE (unsigned ϵ_2), and not the arbitrary sign of ϵ_2 . We therefore chose $|\epsilon_2|$ as a parametric regressor. Ideally, one would include both $|\epsilon_2|$ and ϵ_3 in the same GLM, yet these two trajectories are indeed correlated. We also included discrete regressors with key markers of behaviour at the outcome onset: win, lose, no response.

During this revision process, we have additionally conducted an analysis as suggested by reviewer #3 in which we run the main outcome-locked GLM using regressor ϵ_3 instead of $|\epsilon_2|$. The results, as expected, are very similar, given that both trajectories are correlated (**Supplementary Figure 6**). See our answer to point 3 by reviewer #3.

As in the original manuscript, we separately conducted a GLM in an entirely different time window, the interval locked to the stimulus event (0–1.8 s post-stimulus with statistical analysis of the **0.1–0.7 s** interval). In this time frame, participants make their response (right, left press) and wait for the outcome to be delivered. In this case, the parametric HGF regressors that are hypothesised to modulate neural activity are the prediction regressors ($|\hat{\mu}_2|$, $\hat{\mu}_3$), in addition to discrete, non-HGF regressors such as the stimulus cues, response type (left, right press, no response) and, potentially, outcome events occurring later. Due to collinearity of $|\hat{\mu}_2|$ and $\hat{\mu}_3$, we chose $|\hat{\mu}_2|$ for this model.

- Auksztulewicz R, Friston KJ, Nobre AC. 2017. Task relevance modulates the behavioural and neural effects of sensory predictions. PLoS Biol 15:e2003143. doi:10.1371/journal.pbio.2003143
- Haarsma J, Fletcher PC, Griffin JD, Taverne HJ, Ziauddeen H, Spencer TJ, Miller C, Katthagen T, Goodyer I, Diederer KM, Murray GK. 2021. Precision weighting of cortical unsigned prediction error signals benefits learning, is mediated by dopamine, and is impaired in psychosis. Molecular psychiatry 26(9):5320-33.

16. *Why are alpha and beta analyzed together but independently from gamma (page 26)? Why not make all three independent or analyze all three together? I wonder if this introduces correlations or biases in the analysis. If there are references for this approach, please give them here. If it is due to the physiological dissociation the authors mention in the introduction (page 3), please state this at the point point of the methods where the band-pass filter is mentioned.*

This may not have been clear in the previous version of the manuscript. We have amended the text to clarify that the full range 8–120 Hz was analysed (statistics over 8–100 Hz) for the (new) main GLM using $|\epsilon_2|$, σ_2 and σ_3 as regressors. The convolution model runs from 8 to 120 Hz in consecutive frequency bands of 2 Hz, and the estimates for each bin are independent. For computational reasons, our code did run ranges 8–30 Hz and 32–120 Hz separately but this would be equivalent to running everything in full from 8 Hz to 120 Hz (a computationally expensive process as the SPM code would store the full matrix from 8 to 120 Hz at once).

We hypothesised that the effects of trait anxiety on the neural oscillatory modulation of pwPE would be represented in changes in gamma and alpha-beta activity. We additionally hypothesised that σ_2 and σ_3 would be correlated with neural activity changes in the 8–30 Hz range. This second hypothesis was postulated based on the evidence that precision (inverse uncertainty) is encoded in 8–30 Hz oscillations^{20,30}. However, in the new main outcome-locked GLM, all regressors are convolved with the neural activity for the full range 8–120 Hz.

For the stimulus-locked GLM on the effect of predictions before observing the outcome we analysed exclusively the 8–30 Hz range, based on prior evidence (refs. ^{19,21-25}).

MINOR COMMENTS

Page 4 (bottom, beginning of results section):

- *in the description of the participant samples in the results no specifics of the two groups are given, specifically no group sizes, these are only given for the whole group, and since the total numbers of the participants is odd the matching was probably not 1 to 1.*

Thanks for pointing this out. We have specified this now “Both LTA and HTA samples were matched in age and the proportion of males and females (LTA, $N = 20$, 22.1 [standard error of the mean or SEM, 0.4] years, 12 female; HTA, $N = 19$, 21.7 [0.4] yrs, 12 female)”. We had originally recruited an additional male participant for the HTA sample but their final participation was impaired by Covid-19 lockdown effects. Although we agree that an even and odd sample cannot be exactly matched in sex, they are fairly well balanced (12/8 and 12/7 ratios of female/male participants).

- *could the authors specify in the text how exactly the reward probability trajectory $p(\text{reward}|\text{orange})$ was generated? In the caption to Figure 1B it seems to be that the trajectory was randomized and different for each participant. If this is the case, could the authors please also specify how they defined blocks, or which block lengths were chosen. And was this factored into the comparison between groups*

Please, see our reply to a similar query in question 3 above and the new Supplementary Figure 1.

- *could the authors state before the second to last line (about the heart rate) which physiological quantities were assessed during the experiment?*

Thanks. We have now named the two heart-rate variability measures we used: standard HRV and high-frequency HRV). Details are provided in Methods and Supplementary Methods as we consider that more details on these types of analysis may not be crucial for the reader at this early phase of the results.

Page 6 (figure caption):

- *the abbreviation TB2 is mentioned in the caption but not explained. I assume it is for task block? Same in the x axis label in C and in the Supplementary figure 2.*

See please our reply to a similar query above, question 4.

- *The figure caption speaks about inputs u but u is not shown in the graph of the model.*

Amended. We have removed “ u ” from this caption as this label is not used in the graphic. We refer to inputs as the purple dots in the bottom of panel B.

Page 8: • *There is a bracket missing on the y axis label of Figure 2F.*

Thank you. Amended.

- *Epsilon _{i} does not occur in the text and it is at this point unclear what it is.*

Thank you. Amended. See our reply to question 6.

Page 12: • *please add an x axis label to Figure 3 D,F,H*

We have added a-axis label “Frequency band” to the bottom panel. Similarly to Figure 5.

Page 18: • *in paragraph 2 it should read lose-shift not lose-shit rate :)*

Of course, we apologise. Amended.

Page 25: • can the authors please specify the content of the “custom python scripts” for preprocessing?

The custom python scripts along with the main GLM Matlab/SPM scripts are included in the Open Science Framework project we had created for the study, <https://osf.io/wsjpgk/>. We have specified this now at this point of the revised manuscript (in the earlier version of the manuscript we mention the OSF project later on page 35). In brief, the scripts deal with the standard preprocessing of MEG (50 Hz, 100Hz notch filtering, epoch selection, ICA using FastICA). They also include scripts for source analysis, such as forward modelling, co-registration, running LCMV on single trials and for specific anatomical labels from our ROIs and DKT atlas.

Without knowing the content it is hard to judge if the preprocessing was sound or not, as anything could be done in a custom script.

Agreed. See please <https://osf.io/wsjpgk/>

Reviewer #3

We would like to thank the reviewer for their feedback.

1. *The authors observed enhanced gamma activity and accompanied by suppression of alpha/beta activity during the encoding of pwPEs in HTA. These seems inconsistent with the author's previous study. In Hein et al. (2022), the authors found that state anxiety increased alpha/beta activity in frontal regions in processing of pwPEs. I wonder how to understand this potential inconsistency.*

Thanks for this question. This is one of the key findings. We have tried to made this clearer in the revised manuscript. Our findings demonstrate across both studies a dissociation between the effect of temporary anxiety states and trait anxiety on decision making in this one-armed bandit task: State anxiety primarily attenuated belief updating on level 2, while trait anxiety enhances belief updating on level 2. In the current study, the greater update steps on level 2 in HTA result from increased informational uncertainty (σ_2 or $1/\pi_2$). This is reflected in the form of the update equations of the posterior estimates for level 2:

$$\Delta \mu_2^k = \mu_2^{[k]} - \mu_2^{[k-1]} \propto \frac{1}{\pi_2^{[k]}} \delta_1^{[k]}$$

By contrast, yet consistent with the equation above, in state anxiety we found reduced informational uncertainty and corresponding slower belief updating on level 2. See, for instance, page 21 in the Discussion (“Induced anxiety states, by contrast, ...”).

The best model also differs in each study. In the current study, HTA individuals overestimate the level of volatility in the environment (posterior mean μ_3), and their behaviour is best described by a response model in which decisions are driven by trial-wise volatility estimates. In Hein et al., (2021), where we observed that state anxious individuals

underestimate the environmental volatility, the best model explaining behaviour was the 3-level HGF coupled with a response model where decisions depend on a fixed decision noise parameter.

As we had written in the discussion, the results in state anxiety by Hein et al., (2021) are in line with the outcomes in the “physiological” component of trait anxiety (Wise et al. 2020; Fan et al. 2022).

Given the opposing patterns of behavioural and computational results in state and trait anxiety, we would also expect differences in neural patterns. While enhanced alpha/beta activity was observed in state anxiety during encoding of pwPEs on level 2 (Hein et al., 2022), attenuating belief updates, we now found reduced alpha/beta responses in trait anxiety. In the Bayesian predictive coding framework, downregulation of alpha/beta activity would disinhibit superficial-layer processing of novel information (PE) by increasing gamma (and gamma may reflect population spike activity: Bastos et al., 2020). Consistent with this account, the current study demonstrated enhanced gamma activity in HTA during encoding pwPE on level 2, thereby speeding belief updating about the reward contingencies. We have revised the manuscript to make this clearer. This is included on page 22.

2. *The authors built on recent progress in rhythm-based formulations of Bayesian predictive coding to identify sources of oscillatory modulations associated with altered learning in a volatile environment in subclinical trait anxiety. It is not clear whether a hierarchical model is necessary. Also, the authors did not explore how other types of hierarchical models would perform. If you decide to keep only this hierarchical model, stronger justification is necessary.*

We agree that some of the seminal papers investigating the oscillatory correlates of predictive coding do not use hierarchical models or even Bayesian models to explain behaviour. For instance, the excellent work by Bastos and colleagues (2020, 2014) in monkeys uses repetition of visual stimuli to assess expected and unexpected stimuli as an approximation to predictions and prediction errors (PE). The authors acknowledge that this approach is an “initial tool” yet they explicitly state that in their future work they aim to investigate more complex paradigms and learned associations (Bastos et al., 2020). Other important studies, however, use hierarchical Bayesian models of perception and associate relevant computational variables to oscillatory correlates. One such example is the work by Feldman and Friston (2014) on attention. The authors understand attention as a process that relies on estimating uncertainty during hierarchical inference about the causes of sensory input. Importantly, where previous studies used a hierarchical Bayesian inference model or hierarchical Bayesian predictive coding model to describe behaviour, the investigation of oscillatory activity focused on finding the correlates for specific variables, such as PE or precision or precision-weights, in a specific level of the hierarchy (e.g. Auztulewicz et al., 2017: focused on level 2 of the HGF; Palmer et al., 2019: focused on

level 1 of the HGF). However, the hypothesised mapping between PE or precision-weights and a specific frequency of oscillations applies to all hierarchical levels. In that sense, the reviewer is correct and there are **no strong a priori hypotheses for pwPE on level 2 or 3 being modulated by different oscillatory frequencies**. If pwPEs are encoded by enhanced gamma (and spiking) and reduced alpha/beta, then we would expect this frequency mapping to be expressed across hierarchical levels.

In our study, the rationale for using a hierarchical Bayesian model was two fold: First, previous work suggested that biases in decision making in anxiety could be best addressed by assessing different (hierarchically-related) types of uncertainty while individuals perform a decision-making task with volatility (Bishop and Gagne, 2018; Pulcu and Browning, 2019). We expected that any alteration in Bayesian PC at any hierarchical level of the model would be reflected in the specific frequency rhythms identified in previous work (alpha for precision weights; gamma for pwPE; alpha/beta for predictions).

Second, we were interested in understanding how individuals with high trait anxiety would perform the same volatile task that we used to investigate state anxiety in Hein et al. (2021, 2022). We had specific hypotheses regarding how trait anxiety would modulate those same HGF variables that were identified as relevant to explain state anxiety effects in Hein et al. (2021, 2022). We also had very specific hypotheses regarding how the HGF and behavioural alterations would be expressed in the neural correlates of Bayesian PC.

This was our rationale for using the same computational modelling framework.

In our future work, we are keen to use alternative Bayesian hierarchical models that have been successfully implemented in tasks with volatility (Piray and Daw, 2020; 2021). In particular, the Piray and Daw (2021) proposed a model for joint estimation of volatility and stochasticity (Piray and Daw ; 2021). This model could help identify whether individuals with anxiety “misinterprets noise due to stochasticity as a signal of change, i.e., volatility.”

Direct comparison of the HGF and related models (e.g. the work by Piray and Daw) is however not trivial as model inversion for the HGF uses variational Bayes, while the probabilistic model by Piray and Daw uses Monte Carlo sampling to estimate belief distributions. Reformulating models to the Bayesian inference framework (or a different framework) to allow for model comparison is challenging, yet it has been achieved in previous modelling work, such as in Marković and Kiebel (2016).

An important complementary line of work in mood disorders has primarily focused on using reinforcement learning models (e.g. Gagne et al., 2020; Aylward et al., 2019). However, we were specifically interested in assessing the neural correlates of Bayesian predictive coding, and their modulation by subclinical anxiety. One of the central hypotheses postulated in recent years for anxiety and depression is that symptoms in these conditions are associated with misestimation of different types of uncertainty. Hierarchical Bayesian

models provide estimates on mean and variance (uncertainty) of belief distributions and are therefore most suitable to address hypotheses related to uncertainty and precision-weighted belief updating in psychiatric conditions.

- Marković, D., Kiebel, S.J., 2016. Comparative analysis of behavioral models for adaptive learning in changing environments. *Front. Comput. Neurosci.* 10, 33.
- Bishop SJ, Gagne C. 2018. Anxiety, Depression, and Decision Making: A Computational Perspective. *Annu Rev Neurosci* 41:371–388.
- Pulcu E, Browning M. 2019. The Misestimation of Uncertainty in Affective Disorders. *Trends Cogn Sci.*
- Piray P, Daw ND. 2020. A simple model for learning in volatile environments. *PLoS computational biology* 16(7):e1007963.
- Piray P, Daw ND. 2021. A model for learning based on the joint estimation of stochasticity and volatility. *Nature communications.* 12(1):1-6.

3. *In a Bayesian PC framework, this study revealed that HTA undermines learning through an overestimation of volatility leading to faster belief updating. The authors found the HTA individuals were significantly more uncertain about the environment. However, it's incomprehensible to me, no significant differences were found in uncertainty about volatility or the tonic learning rates at levels. In my opinion, the authors should explain these insignificant results in the discussion section.*

This is an interesting aspect which we now revisit and briefly discuss in the Results (p.11-12) and Discussion sections (p.21,23). Concerning volatility, we observed that HTA had a higher expectation on volatility than LTA (posterior mean of the belief distribution on level 3: μ_3). HTA also had greater environmental uncertainty, which is defined by Mathys and colleagues (2014) as: $\exp(\kappa\mu_3^{(k-1)} + \omega_2)$. This variable also depends on the tonic volatility, ω_2 . In this context, μ_3 and environmental volatility are closely related, despite ω_2 playing an additional role in the modulation of environmental uncertainty. Previous work suggested that higher levels of estimated environmental uncertainty or expected volatility may inflate the degree to which new outcomes update beliefs, increasing learning rates (Lawson et al., 2017; Jepma & Murphy, 2016; Piray & Daw, 2021), as we observed here.

Uncertainty about volatility, on the other hand, corresponds to parameter σ_3 . This is the posterior variance of the belief distribution on level 3. This parameter reflects imperfect knowledge about the estimate μ_3 , similarly to informational uncertainty on level 2 (σ_2).

Experimental and control groups could exhibit differences in their estimated level of environmental volatility (posterior mean, μ_3) yet be similarly uncertain (precise) about those estimates (posterior variance, σ_3). Our results indeed reveal that HTA participants had a generally higher expectation on volatility than LTA, yet both groups did not differ in their uncertainty in those estimates. We have not included the results of a new between-group statistical analysis of the initial belief on volatility ($\mu_3^{(0)}$). The initial prior on volatility $\mu_3^{(0)}$ is a free parameter in the winning model (response model from Diaconescu et al., 2014). In the revised manuscript, we updated the results: “HTA individuals had a greater initial estimate on volatility (free parameter $\mu_3^{(0)}$) than LTA participants ($P_{FDR} = 0.024 < 0.05$, $\Delta = 0.72$, $CI =$

[0.54, 0.87]; Figure 2E). Over trials, we observed that posterior mean on log-volatility estimates, μ_3 , remained higher in the HTA group relative to the LTA group ($P_{FDR} = 0.019 < 0.05$, $\Delta = 0.74$, $CI = [0.55, 0.87]$; Figure 2F).”

Then, after reporting the non-significant differences in σ_3 , we add: “The latter outcome suggests that trait anxiety had no significant effect on the speed of updates about volatility (σ_3 weights prediction errors updating level 3). Rather, trait anxiety led individuals to overestimate the level of volatility in the environment already from the start, and this estimate remained high throughout the task.”

On the other hand, the tonic learning rates, ω_2 and ω_3 , can be considered to be evolution rates defining the relationship between levels. Previous work comparing induced conditioned hallucinations in psychosis patients and healthy controls using HGF models reported differences in volatility estimates (among other effects, e.g. in μ_2 , μ_1) but no effects in ω_2 or ω_3 (Powers et al., 2017). Thus, the findings of Powers and colleagues support that group differences in the posterior mean of belief trajectories could emerge even if the coupling between levels is not different between groups.

Our results demonstrate that HTA and LTA individuals primarily diverged in their inferred level of environmental volatility (initially and throughout the task). This overestimation effect cascaded down the HGF hierarchy, influencing how individuals updated their beliefs on level 2 (faster: HTA / slower: LTA). Yet the coupling between levels was not different between groups in our study.

- Powers AR, Mathys C, Corlett PR. 2017. Pavlovian conditioning–induced hallucinations result from overweighting of perceptual priors. *Science* 357:596–600.
 - Lawson RP, Mathys C, Rees G. 2017. Adults with autism overestimate the volatility of the sensory environment. *Nat Neurosci* 20:1293–1299. doi:10.1038/nn.4615
 - Jepma M, Murphy PR, Nassar MR, Rangel-Gomez M, Meeter M, Nieuwenhuis S. 2016. Catecholaminergic Regulation of Learning Rate in a Dynamic Environment. *PLoS Comput Biol* 12:e1005171. doi:10.1371/journal.pcbi.1005171
 - Piray P, Daw ND. 2021. A model for learning based on the joint estimation of stochasticity and volatility. *Nature communications*. 12(1):1-6.
 - Mathys CD, Lomakina EI, Daunizeau J, Iglesias S, Brodersen KH, Friston KJ, Stephan KE. 2014. Uncertainty in perception and the Hierarchical Gaussian Filter. *Front Hum Neurosci* 8:825. doi:10.3389/fnhum.2014.00825
4. *Also, the authors employed a three-level hierarchical Bayesian model to test the rhythm-based formulations of Bayesian predictive coding, and revealed that the significant difference mainly comes from the third level. However, it's strange to only focus on the encoding of pwPEs at level 2 (e.g., unsigned precision-weighted prediction errors about stimulus outcomes, $|\varepsilon_2|$). What are the results at level 3 (e.g., $\varepsilon_3, \sigma_3, \mu_3$)?*

Due to collinearity of regressors, we had chosen $|\varepsilon_2|$ over ε_3 primarily for theoretical reasons: The overestimation of volatility μ_3 in HTA leads to more stochastic decisions and faster belief updates on level 2. We therefore hypothesised that HTA individuals would exhibit neural responses correlated with greater changes updating their beliefs on the

tendency of the reward contingencies. This is the analysis we conducted, which also allowed us to interpret the neural results in trait anxiety in light of our previous results in state anxiety in Hein et al., 2021. Note that in the revised manuscript, following a query by reviewer #2, we include $|\varepsilon_2|$, σ_2 and σ_3 as parametric regressors in the main outcome-locked GLM. This is described in detail in page 13. Excerpt:

*“The main outcome-locked GLM evaluated the effect of parametric regressors $|\varepsilon_2|$, σ_2 , and σ_3 , on the TF responses, while it controlled for the effect of discrete outcome events (win, lose, no response). Next, in an exploratory analysis, we implemented a stimulus-locked GLM to assess the neural oscillatory processes correlated with the parametric regressor $|\hat{\mu}_2|$. This model additionally included discrete regressors denoting the stimuli presentation (blue image on the left or right side), the participant’s response (left, right, no response), and outcome cues (win, lose, no response) at their respective onset (**Methods**).”*

In the new main GLM, the neural responses to $|\varepsilon_2|$ and σ_2 are very similar to the effects we reported in the original manuscript.

Assessing the neural correlates of ε_3 updating level 3 can also be interesting approach. Ideally, we would have preferred to use both trajectories $|\varepsilon_2|$ and ε_3 for GLM analysis, yet such a model would not be properly defined due to collinearity. We hope to explore other hierarchical models in future work and assess their suitability for GLM analyses.

Following reviewer #3’s suggestion, we have conducted the (new) main convolution model by replacing $|\varepsilon_2|$ with ε_3 to assess the correlated outcome-locked activity. The results of the GLM analysis on ε_3 are displayed below and presented in **Supplementary Figure 6**:

A visual comparison of the same GLM using either $|\varepsilon_2|$ (**Figure 3**) and ε_3 highlights the similarity of the results, as expected, given that these trajectories correlate.

We have also created for reviewer #3 a figure to illustrate the group-average time course of the posterior mean of the beliefs on volatility (panel A below) and on the tendency of the reward contingencies (panel B). Panel A illustrates the sustained higher expectation on in HTA. Note that the group-average of is not particularly meaningful as this depends on the underlying contingencies, which differ in each individual (see reply to point 3 by reviewer #2, the new **Supplementary Figure 1**, and pages 4-5 in the revised manuscript).

Reviewers' comments:

Reviewer #2 (Remarks to the Author):

Firstly, I would like to thank the authors for revising their manuscript, incorporating the comments, running new analyses and adding more detail in regards to the experimental design.

The additional details have however raised some questions in regards to their behavioral and modeling results.

The authors pseudorandomly generated action-outcome contingency sequences for each participant separately. They then analyzed behavioral findings like the win rate, and also ran inference on the behavioral data with a HGF model. Given that each contingency sequence for each participant was different, I think the results may unfortunately be confounded by the task design. I don't think the statistical test in Supplementary Figure 1A is enough to ensure that there are no differences in trajectories that influence behavior and subsequent analyses.

Two main quantities affect participant choice behavior and are also inferred using the behavioral model: the uncertainty and volatility. Since each participant saw a different reward contingency trajectory, it would be important to ensure that the uncertainty and volatility in these trajectories are the same for each participant, or at least between groups, since groups are being compared with respect to their estimates of these quantities. Same holds for the comparison between blocks of the experiment.

Through visual inspection of these trajectories plotted in Supplementary Figure 1, it seems that both, experienced uncertainty and volatility, may actually differ between groups.

- Volatility (shown in Supplementary Figure 1B): Two main differences are visible here.
 - o The pseudorandom script did apparently not ensure a contingency switch in between blocks of the experiment. As a consequence, not all participants experienced the same amount of switches in the experiment, and less switches lead to less volatility in the experiment. Concretely, in the LTA group, 5 subjects experienced one switch less, and 3 subjects in the HTA group. This suggests that the HTA group overall experienced more volatility by task design. Unfortunately, it is also a main finding in Figure 2F that HTA estimate volatility to be larger. I am however not sure, if this conclusion can be drawn confidently, given that the group also experienced more volatility.
 - o It looks like the amount of trials before the first switch differs between groups, where on average LTA subjects experienced a longer stable phase before the first switch compared to the HTA group. In Figure 2E it is a main result that HTA estimate more initial volatility, which however may stem from the fact that they on average experienced more volatility (earlier switches) early on. Hence I am also not sure if this conclusion can be drawn confidently. The relatively increased early volatility may also bias their estimate of the overall volatility in a way that is not captured by the model.
- Uncertainty (Supplementary Figure 1A): The authors show in this figure the average trajectories of each group. It is visible that the HTA group starts on average with more uncertainty ($p(\text{win})$ closer to 0.5). Additionally, it seems like the overall trajectory of the HTA group is closer to 0.5 than the LTA group, which would mean that the HTA group was subjected to more uncertainty in the task. This could be due to the fact that the stable lengths of the contingencies were randomized and differently long for different subjects, so that some subjects could have 20 trials more with 50/50 than others, and so on. It is a main result in Figure 2 that HTA subject estimate uncertainty higher. However, if the task by design subjected HTA individuals to more uncertainty, this conclusion can also potentially not be drawn confidently. Furthermore, in Figure 1 the win rates of the two groups are shown. It is not clear whether the two groups would have been able to achieve the same win rate, even if both behaved optimally, due to the increased reward uncertainty in the HTA group. Hence I am also not sure if this conclusion can be made confidently.

Unfortunately, the inferred uncertainty and volatility parameters of the model are also a key factor for

the MEG analysis, therefore, if the task design does in fact confound the behavioral results, it may also confound the neuroimaging results.

This overall may be a strong limitation of the work.

In order to test the influence on behavior and model fitting, the authors can calculate the uncertainties and volatilities of the reward trajectories and see if these correlate with the inferred model parameters.

- I think it would be more informative if the authors plotted the average uncertainty rather than the average contingency in Supplementary Figure 1A, by plotting $p(\text{win}|\text{best option}) - 0.5$ instead.
- One can calculate the uncertainty and volatility of each trajectory, and check if there are group differences. To ensure that the inferred model parameters are not biased by the task design, the authors could check if the experienced and inferred volatilities and uncertainties correlate. To make more reliable claims about group differences in that regard, the authors could regress these trajectory parameters out of their group comparison.

I sincerely hope that the effect of the task design is negligible, then the MEG analyses would hold unchanged. If there are confounds due to the task design, one would have to see whether the MEG analyses can be adjusted to correct for this or not.

In any case, the authors should discuss this as a limitation of the study.

New minor comments:

1. Page 3 first paragraph. "... PC is thought orchestrated by ..." should probably be "... thought to be orchestrated by ...".
2. Page 4: The task should technically be a two-armed bandit. Even if the reward probabilities are inverse, there are still two "lotteries" participants can play. However, I don't know whether this has to be said at all, you could also just say it is similar to a reversal learning task. This would help the reader at that point too.
3. Please mention in the caption of Fig 1 B that this is an example, for example something like "The probability governing the likelihood of the orange stimulus being rewarded, $p(\text{win}|\text{orange})$, for one example participant".
4. Page 9 bottom paragraph reads as "higher log-volatility leads to lower decision noise, leads to noisier mapping between beliefs and responses." Shouldn't this be the other way round?
5. Page 8/9 I would introduce kappa and omega around here already.

In regards to the previous comments, I am going to save space and not copy paste the full exchange here again. I think the authors did really well in making their manuscript a lot more clearer in many aspects. And the analyses are now very comprehensible. I would like to thank the authors for incorporating the feedback and working on all of the points.

Reviewer #3 (Remarks to the Author):

Authors have addressed all of my concern.

Reviewers' comments:

Reviewer #2 (Remarks to the Author):

Firstly, I would like to thank the authors for revising their manuscript, incorporating the comments, running new analyses and adding more detail in regards to the experimental design.

The additional details have however raised some questions in regards to their behavioral and modeling results.

The authors pseudorandomly generated action-outcome contingency sequences for each participant separately. They then analyzed behavioral findings like the win rate, and also ran inference on the behavioral data with a HGF model. Given that each contingency sequence for each participant was different, I think the results may unfortunately be confounded by the task design. I don't think the statistical test in Supplementary Figure 1A is enough to ensure that there are no differences in trajectories that influence behavior and subsequent analyses.

Two main quantities affect participant choice behavior and are also inferred using the behavioral model: the uncertainty and volatility. Since each participant saw a different reward contingency trajectory, it would be important to ensure that the uncertainty and volatility in these trajectories are the same for each participant, or at least between groups, since groups are being compared with respect to their estimates of these quantities. Same holds for the comparison between blocks of the experiment.

This is an important point. Thanks for pointing this out. We have conducted validation analyses as requested by the reviewer. Using Bayesian statistics, we provide evidence that the trial-wise stimulus-outcome contingency mapping (commensurate with "reward uncertainty") and true experienced volatility are the same in both groups. In addition, we replicate the main between-group results of the manuscript on behavioural, computational and neural variables using two subsamples of 16 HTA and 16 LTA participants who experienced the exact same number of contingency switches overall (9). We provide details in the following.

The first revised version (R1) of this manuscript indicated that each participant observed a different combination of five phases of stimulus-outcome probability mappings in each block. The possible mappings in each block were 0.9/0.1, 0.7/0.3, 0.5/0.5, 0.3/0.7, 0.1/0.9. As the reviewer pointed out, the task script did pseudorandomly generate the order of these phases independently in each block and participant. Accordingly, it was possible that the same probability mapping would occur at the end of block 1 and, after a break, at the beginning of block 2.

Supplementary Figure 1 (R1) revealed the cases in which this happened. There is a *caveat*, however, which we apologise for: while conducting the control analyses for this second revision (R2), we noticed that the first LTA trajectory in Supplementary Figure 1B of revision R1 was identical to the bottom LTA trajectory. We highlight these duplicated contingency mapping trajectories here (in maroon):

We checked the plot script and found an error in the concatenation of data for the plots A and B. We have reloaded the behavioural data and generated the correct **Supplementary Figure 1** for R2. The new plot can be obtained using the individual participant data from the open science framework repository (OSF, uploaded on 25th July 2022: <https://osf.io/xsjm5/>).

We have redrawn **Supplementary Figure 1** for the new revised manuscript (R2). Note that we now swap A and B, as the panel B represents the group-average of the individual traces in A. We mention this in the updated figure caption (in bold):

Supplementary Figure 1. A) Individual time courses of stimulus-outcome probabilistic relationships for individuals in the **high trait anxiety (HTA, orange) and low trait anxiety (LTA, purple) groups.** **B)** The probability governing the likelihood of the blue stimulus being rewarded, $\text{Pr}(\text{win} | \text{blue})$, across 320 trials is displayed as mean (and SEM) within each group. **The mean traces in (B) represent the group-average of the individual traces in (A).** There were no systematic (or significant) differences between HTA and LTA in the contingency mapping values ($P > 0.05$). **Furthermore, Bayesian statistical analysis provided strong evidence that the two population means in this quantity were equal (Supplementary Results).**

Visual inspection of the amended **Supplementary Figure 1A** reveals that *three* HTA participants and *four* LTA participants did not experience a change in contingency mapping from block 1 to block 2. On the other hand, 16/19 HTA and 16/20 LTA participants experienced a contingency mapping change from block 1 to block 2.

In **Supplementary Figure 1B** we observe that the group-average $P(\text{win}|\text{blue})$ trajectory closely overlaps in both groups throughout the task. Visual inspection suggests a potential group difference around trials 90-95, such that LTA individuals have a higher average $\text{Pr}(\text{win}|\text{blue})$ in this phase, while for HTA participants this trial range would have a higher $\text{Pr}(\text{win}|\text{orange})$.

- I think it would be more informative if the authors plotted the average uncertainty rather than the average contingency in Supplementary Figure 1A, by plotting $p(\text{win}|\text{best option}) - 0.5$ instead.

Done. We plot the same panel below after subtracting 0.5 as the reviewer requested.

During trials 90-95, we can see that each group has a larger average contingency mapping towards one of the two images (blue for LTA and orange for HTA), but both groups show similar absolute deviations from 0. And both groups have a very similar average throughout both blocks.

A statistical analysis of between-group differences in $\text{Pr}(\text{win}|\text{blue})$ using permutation tests revealed no effect (all uncorrected p-values were above 0.05, except in trials 91-92, where the p-value was 0.0418; Applying FDR control for multiple comparisons further supports the lack of a group effect on $\text{Pr}[\text{win}|\text{blue}]$; $P_{\text{FDR}} > 0.05$).

See below the trajectory of uncorrected p-values for between-group differences in $\text{Pr}(\text{win}|\text{blue})$ (the red line denotes uncorrected alpha level 0.05):

In the new revision, as recommended by the reviewer, we have conducted several validation analyses of our group effects. Using Bayesian statistics, we provide evidence that both groups experienced the same true volatility. Our analyses also show that both groups were exposed to the same probability of stimulus-outcome relationship, $\text{Pr}(\text{win} | \text{blue})$: the probability that the blue stimulus is rewarding on a given trial. This quantity is commensurate with “reward uncertainty”, as the reviewer mentions: $\text{Pr}(\text{win} | \text{blue}) - 0.5$.

All participants experienced five phases in each block with $\text{Pr}(\text{win} | \text{blue})$ taking values 0.9, 0.1, 0.5, 0.3, 0.7 (in a pseudorandomised order), which changed every 26-38 trials over the course of 160 trials in each block. The maximum “reward uncertainty”, as denoted by the reviewer, happens when participants experience a contingency mapping of 0.5. This happened once in block 1 and once in block 2 in all participants.

The group-average $\text{Pr}(\text{win} | \text{blue})$ shown in the new **Supplementary Figure 1B** displays whether the ordering of the contingency phases had some **bias** in one group, which could lead to confounding effects. For instance, it could be that by chance every LTA participant experienced a 0.5 contingency mapping in the first trials of the experiment, while all HTA participants could have experienced a 0.9 contingency mapping at the beginning. We show in the **Supplementary Figure 1B** and in our validation analysis below that this is not the case.

Validation analyses:

Given that we aim to find evidence for the null hypothesis that there are no differences between groups in true experienced volatility or in $\text{Pr}(\text{win} | \text{blue})$ (related to “reward uncertainty”), we estimate the Bayes Factor in each analysis. Next, we conduct additional validation analyses by comparing the main dependent variables between the subsets of participants who experienced exactly nine contingency mapping changes throughout the task (16/19 HTA and 16/20 LTA; excluding participants who experienced eight switches).

See below.

Through visual inspection of these trajectories plotted in Supplementary Figure 1, it seems that both, experienced uncertainty and volatility, may actually differ between groups. One can calculate the uncertainty and volatility of each trajectory, and check if there are group differences.

- Volatility (shown in Supplementary Figure 1B – **now 1A**): Two main differences are visible here.
 - o The pseudorandom script did apparently not ensure a contingency switch in between blocks of the experiment. As a consequence, **not all participants experienced the same amount of switches in the experiment, and less switches lead to less volatility in the experiment**. Concretely, in the LTA group, 5 subjects experienced one switch less, and 3 subjects in the HTA group. This suggests that the HTA group overall experienced more volatility by task design. Unfortunately, it is also a main finding in Figure 2F that HTA estimate volatility to be larger. I am however not sure, if this conclusion can be drawn confidently, given that the group also experienced more volatility.
 - o It looks like the amount of trials before the first switch differs between groups, where on average LTA subjects experienced a longer stable phase before the first switch compared to the HTA group. In Figure 2E it is a main result that HTA estimate more initial volatility, which however may stem from the fact that they on average experienced more volatility (earlier switches) early on. Hence I am also not sure if this conclusion can be drawn confidently. The relatively increased early volatility may also bias their estimate of the overall volatility in a way that is not captured by the model.

Visual inspection of the amended panel A in **Supplementary Figure 1** reveals that *three* HTA participants and *four* LTA participants did not experience a change in contingency mapping from block 1 to block 2. On the other side, 16/19 HTA and 16/20 LTA participants experienced a contingency mapping change from block 1 to block 2.

Bayes factor analysis.

We evaluated between-group differences by computing Bayes Factors (BF) using the bayesFactor toolbox (<https://github.com/klabhub/bayesFactor>) in MATLAB. This toolbox implements tests that are based on multivariate generalisations of Cauchy priors on standardised effects (Rouder et al., 2012).

For between-group comparisons of a dependent variable (DV), we calculated the BF on the model $DV \sim 1 + \text{group}$, where DV is explained by a fixed effect of group (HTA, LTA). The model was fitted using the fitlme function of the MATLAB Statistics toolbox. Computing BF allowed us to quantify the evidence in support of the alternative hypothesis (full model, in our case assessing the main effect of the group) relative to the null model (intercept-only model, i.e., $DV \sim 1$).

This approach was implemented using measures of the true experienced volatility as DV.

On the other hand, for a factorial analysis, such as the effect of group and trial bins on $\Pr(\text{win} | \text{blue})$, we used as full model $DV \sim 1 + \text{group} * \text{bin}$, which includes three categorical fixed effects: group, bin and interaction group:bin. Next, we constructed the restricted models by excluding each of the main or interaction effects. Last, we computed the ratio of the full model and each restricted model. The resulting BF provided evidence for either main effect (group, bin) or interaction effect. Here, we transformed the 320 trials into a categorical variable, bins of 32 trials. Thus, we assessed the BF of a 10 x 2 ANOVA model with factors bin (10 levels: average within each bin of 32 trials) and group (HTA, LTA).

BF values were interpreted as in Andraszewicz et al. (2015). As BF is the ratio between the probability of the data being observed under the alternative hypothesis and the probability of the same data under the null hypothesis,

$$BF_{10} = \text{likelihood of data given H1} / \text{likelihood of data given H0}$$

a BF_{10} of 20 would indicate strong evidence for the alternative hypothesis. On the other hand, BF of 0.05 would provide strong evidence for the null hypothesis (see Table 1 by Andraszewicz et al., 2015 for further details).

We have estimated the following quantities to reflect the true experienced volatility by each participant:

- (a) **Number of switches in contingency mapping:** this is 9 in all participants except for 3 HTA and 4 LTA participants, who have 8 switches respectively. There is moderate evidence for the null hypothesis, supporting that this quantity is equal in both groups: $BF_{10} = 0.21$ (Bayes factor in range $1/3 - 1/10$ is associated with moderate evidence for H_0). The p-value obtained using between-group permutation tests is $P = 0.6943$. Descriptive statistics indicate that HTA participants had 8.84 (SEM 0.09) switches over 320 trials, while LTA participants had 8.80 (SEM 0.09)
- (b) **Average number of trials after which participants experience a change in the contingency mapping:** 32.80 (0.42) for HTA, 32.76 (0.49) for LTA. There is moderate evidence in support of the null hypothesis, $BF_{10} = 0.2057$ ($P = 0.9670$).

- **Uncertainty (Supplementary Figure 1A - now 1B):** The authors show in this figure the average trajectories of each group. It is visible that the HTA group starts on average with more uncertainty ($p(\text{win})$ closer to 0.5). Additionally, it seems like the overall trajectory of the HTA group is closer to 0.5 than the LTA group, which would mean that the HTA group was subjected to more uncertainty in the task. This could be due to the fact that the stable lengths of the contingencies were randomized and differently long for different subjects, so that some subjects could have 20 trials more with 50/50 than others, and so on. It is a main result in Figure 2 that HTA subject estimate uncertainty higher. However, if the task by design subjected HTA individuals to more uncertainty, this conclusion can also potentially not be drawn confidently.

This is not the case in the amended **Supplementary Figure 1**. See our analysis at the beginning of this response letter. Both groups had on average a very similar and overlapping trajectory of contingency changes, despite the phases being individually pseudorandomised (uncorrected trial-wise $P > 0.05$, except for trial 91 and 92; after control of the FDR, there is no group effect in the average $Pw(\text{blue} | \text{win})$, $P_{\text{FDR}} < 0.05$). Thus, the group did not experience different “reward uncertainty”.

Computing BFs for the bin x group ANOVA analysis of the $Pr(\text{blue} | \text{win})$ trajectory, we obtain the following:

BF for main effect of bin = $6.7482e-04$, demonstrating extreme evidence for H_0 , supporting that the average $Pw(\text{blue} | \text{win})$ was not modulated across bins.

BF for main effect of group = 0.0655, providing strong evidence in favour of H_0 that the group factor does not modulate the population means. Thus, the population mean in both groups was equal.

BF for interaction effect = 0.0015 Extreme evidence for a lack of interaction effect.

Last, as a sanity check, we estimated BF separately on trials 91-92, assessing between-group differences with an independent two-sample t-test model. We obtained $BF_{10} \sim 1.7$, which demonstrates *anecdotal* evidence for the alternative H_1 in trials 91-92.

BF_{10} for the remaining trials was close to 0.2, reflecting moderate evidence in favour of H_0 , and thus suggesting that the population means on each trial was equal.

Our analyses above demonstrate that, although we pseudorandomised the contingency phases separately in each individual, and 3/19 HTA and 4/20 LTA participants did not experience a switch from block 1 to 2, this **did not contribute to group differences in true experienced volatility or the trial-wise stimulus-outcome probabilistic relationship**.

1. Rouder JN, Morey RD, Speckman PL, Province JM (2012) Default Bayes factors for ANOVA designs. *J Math Psychol* 56:356–374.
2. Andraszewicz S, Scheibehenne B, Rieskamp J, Grasman R, Verhagen J, Wagenmakers E-J (2015) An Introduction to Bayesian Hypothesis Testing for Management Research. *J Manag* 41:521–543.

- To ensure that the inferred model parameters are not biased by the task design, the authors could check if the experienced and inferred volatilities and uncertainties correlate.

We have assessed for the reviewer whether the inferred volatility estimate correlates with the true experienced volatility and whether this could account for group differences. We have chosen a non-parametric Spearman rank correlation instead of Pearson correlation because the “true volatility” measures and the inferred volatility estimates are on very different scales and units and could be associated in a non-parametric way.

Specifically, we conducted a simple correlation across all 39 participants between the individual log-volatility (μ_3) estimate (collapsed across trials) and the true (experienced) volatility, which is the rate of how often the contingency changes over the 320 trials. This is quantity (a) or (b) above. A Spearman rank correlation between the number of experienced switches ([a]) and the inferred volatility estimate (mean μ_3 trajectory) provided a Spearman ρ close to zero, $\rho = 0.0891$ ($P = 0.5898$).

A Spearman rank correlation between the number of trials after which a participant experiences a switch([b]) and μ_3 also provided a Spearman ρ close to zero, $\rho = -0.012$ ($P = 0.9941$).

- To make more reliable claims about group differences in that regard, the authors could regress these trajectory parameters out of their group comparison.

We have effectively regressed these parameters out by conducting between-group analyses on the main behavioural, computational and neural DVs in the subsamples of 16 HTA and 16 LTA participants who experienced an identical number of switches in contingency mappings (9; excluding those who experienced a repeated probabilistic mapping from block 1 to block 2):

Reanalysis of computational variables in **Figure 2**:

The figure above is similar to **Figure 2** panels E-H but represents the group analysis with the subsamples of 16 HTA and 16 LTA participants observing 9 switches in probabilistic mapping. The participants who observed 8 switches are marked as “excluded” and denoted by the crosses. Visual inspection of the excluded participants in panels A and B indicates that some participants had a large expectation on log-volatility despite being exposed to slightly smaller true volatility (8 switches instead of 9).

Between-group statistical analysis in the 16-16 subsamples demonstrated that:

- (A) HTA individuals (red) had a greater initial expectation or prior on log-volatility than LTA (purple, $P_{\text{FDR}} = 0.0454 < 0.05$, $\Delta = 0.70$, $\text{CI} = [0.52, 0.87]$; group effects denoted by the black line at the bottom).
- (B) Over time, the posterior mean on log-volatility (μ_3) in HTA remained significantly higher relative to LTA ($P_{\text{FDR}} = 0.0378 < 0.05$, $\Delta = 0.72$, $\text{CI} = [0.52, 0.88]$).
- (C) Informational (estimation) belief uncertainty about the stimulus outcome tendency was greater in HTA compared with LTA ($P_{\text{FDR}} = 0.0012 < 0.05$, $\Delta = 0.81$, $\text{CI} = [0.62, 0.92]$).
- (E) The HTA individuals were also significantly more uncertain about the environment ($P_{\text{FDR}} = 0.0008 < 0.05$, $\Delta = 0.82$, $\text{CI} = [0.65, 0.92]$).

- Furthermore, in Figure 1 the win rates of the two groups are shown. It is not clear whether the two groups would have been able to achieve the same win rate, even if both behaved optimally, due to the increased reward uncertainty in the HTA group. Hence I am also not sure if this conclusion can be made confidently.

Using the HTA and LTA subsamples, we also replicated the result that HTA had a smaller **win rate** in the first block ($P = 0.0370$, non-parametric effect size $\Delta = 0.7031$) but not in the second one ($P = 0.9181$, $\Delta = 0.5039$), when compared to LTA.

Accordingly, when excluding participants with 8 instead of 9 contingency mapping changes, we replicate the main behavioural and computational results of the study (note however the smaller sample sizes). Crucially, HTA participants in the total sample and subsample had a greater expectation on log-volatility, primarily due to an initially higher estimate. Thus, the switch in contingencies from block 1 to block 2 has negligible effects on the inferred volatility estimate in the task. It is also important to note that HTA individuals in the total sample and subsample had a greater expectation on informational belief uncertainty (σ_2). Accordingly, they update their beliefs about the tendency of the stimulus-outcome contingencies faster, using greater steps. This computational result is an important driver of the neural oscillatory effects on pwPE, which we also replicate in the subsamples:

This figure is similar to Figure 3, panels C, E, G but using the 16 HTA and 16 LTA subsamples. The black and white contours denote between-group effects with cluster-based permutation testing, FWER-controlled. We replicate the same group effects in gamma and alpha-beta frequency ranges and in the same ROIs.

- Unfortunately, the inferred uncertainty and volatility parameters of the model are also a key factor for the MEG analysis, therefore, if the task design does in fact confound the behavioral results, it may also confound the neuroimaging results.
- I sincerely hope that the effect of the task design is negligible, then the MEG analyses would hold unchanged. If there are confounds due to the task design, one would have to see whether the MEG analyses can be adjusted to correct for this or not.

Overall, we provide strong evidence that there were no group differences in the stimulus-outcome probabilistic relationships, and also moderate evidence in support of the true experience volatility having equal means in both groups. We also show that the inferred volatility estimate was not correlated with the true experienced volatility (at least to the level of subtle differences from 9 to 8 switches in our experiment). The analyses in the HTA and LTA subsamples replicate the main behavioural, computational and neural effects. Accordingly, the validation analyses combined support our claim that the study findings are robust and are not confounded by individual changes in the pseudorandomised order of contingency mappings.

We will certainly change the task script in future work to make sure that such validation analyses are not necessary and to guarantee that every individual experiences the exact same number of contingency changes. However, the pseudorandomisation of the contingency mapping phases introduced only subtle individual differences which had a negligible effect in our group results.

We sincerely thank the reviewer for their very thorough check of the different manuscript versions and for raising this point, which we have now addressed.

In the R2 revised manuscript, we present the new control analyses with Bayes Factors and on the 16/16 subsamples in **Supplementary Materials** ("Validation analyses: Effect of pseudorandomised order of contingency mappings") and mentioned them in the **Results** section, page 5:

"There were no systematic differences between groups in the order of contingency mappings (Supplementary Figure 1). In 4/20 LTA and 3/19 HTA participants, however, the probabilistic mapping did not change from block 1 to 2, and thus these participants encountered a total of eight contingency mapping changes across the 320 trials, while 16/20 LTA and 16/19 HTA individuals encountered nine probabilistic changes overall. Control analyses provided strong evidence in support of the null hypothesis that both groups were exposed to the same probabilistic mapping over time. There was also moderate evidence that both groups experienced on average an equal amount of true volatility (Supplementary Results: Validation analyses). Additional control analyses further supported that the main behavioural and computational group results were not confounded by individual differences in the

pseudorandomised order of contingency mappings (Supplementary Results: Validation analyses).”

Also, on page 4:

“The stimulus-outcome contingency mapping changed **four times across the 160 trials in each block (every 26-38 trials)**, and the **five** possible contingencies **each block** were 0.9/0.1, 0.1/0.9, 0.7/0.3, 0.3/0.7, and 0.5/0.5,”

New minor comments:

1. Page 3 first paragraph. “... PC is thought orchestrated by ...” should probably be “... thought to be orchestrated by ...”.

Done.

2. Page 4: The task should technically be a two-armed bandit. Even if the reward probabilities are inverse, there are still two “lotteries” participants can play. However, I don’t know whether this has to be said at all, you could also just say it is similar to a reversal learning task. This would help the reader at that point too.

Agreed. We have simply stated “reversal learning task”.

3. Please mention in the caption of Fig 1 B that this is an example, for example something like “The probability governing the likelihood of the orange stimulus being rewarded, $p(\text{win} | \text{orange})$, for one example participant”.

Done. We have added “example” to the caption. We have also added “See individual traces of contingency changes in **Supplementary Figure 1A**. “

4. Page 9 bottom paragraph reads as “higher log-volatility leads to lower decision noise, leads to noisier mapping between beliefs and responses.” Shouldn’t this be the other way round?

Yes, thanks. Initially we had used the term “inverse decision noise”. The larger this parameter, the more deterministic the decisions are (the sigmoid approaches a step function) and thus less noisy. Some authors call it “decision noise” parameter (Mathys et al., 2014; Diaconescu et al., 2014), which we adopted in revision R1. Quoting from Chris Mathys’ Dissertation, “Decision noise levels were chosen in a range from very high (0.5) to very low (24).” Thus, they refer to “a type” of decision noise parameter, which inversely scales with noise in the decisions. For simplicity, we have now used the original term, “inverse decision noise”.

5. Page 8/9 I would introduce kappa and omega around here already.

Done. We introduce the coupling equation (2) right after (1) on page 9.

In regards to the previous comments, I am going to save space and not copy paste the full exchange here again. I think the authors did really well in making their manuscript a lot more clearer in many aspects. And the analyses are now very comprehensible. I would like to thank the authors for incorporating the feedback and working on all of the points.

Thank you.

Reviewer #3 (Remarks to the Author):

Authors have addressed all of my concern.

Thank you.

REVIEWERS' COMMENTS:

Reviewer #2 (Remarks to the Author):

I thank the authors for the very extensive validation analysis. I am happy to say that I am now fully convinced by the results and have no further comments or questions.